# "Amphions Harp gaue sence vnto stone Walles": The Five Senses and Musical–Visual Affect

Katie Bank 

Department of History, University of Birmingham, Edgbaston B152TT, UK; k.n.bank@bham.ac.uk

**Abstract:** In 1582 George Whetstone described the feeling of entering a barren Great Chamber the morning after a night of sparkling social and musical entertainments. Recounting the previous night's activities, he reflected on the relationship between musical activity and space, saying 'the Poets fayned not without reason, that Amphions Harp gaue sence vnto stone Walles'. This article explores the complex relationships between sensing, sociability, activity, and space through an in-depth examination of a late sixteenth- and seventeenth-century English interior design trend: personifications of the Five Senses. Using active imagining, it considers how the Five Senses engaged early modern English subjects in a dialectic between sensory/bodily absence and presence as a mode for exploring the precarious pleasure of holding the passions on the edge of balance. Looking at the spatial and musical-ritual framings of the Five Senses decoration at Knole House, Kent, it investigates how feeling, sensing bodies experienced musical–visual sensory interplay in early modern elite households. It seeks to better understand the aesthetic, emotional experiences of those who gave life to the musical, social situations in such spaces.

**Keywords:** Tudor music; Jacobean music; Five Senses; Knole House; visual culture

In 1582 George Whetstone published *An Heptameron of Civil Discourses* in which he, through a fictitious scenario, described the feeling of entering a barren Great Chamber the morning after a night of sparkling social and musical entertainments:[1]

> I entred the great Chamber, with as strange a regarde, as he that commeth out of a House full of Torch and Taperlights, into a darke and obscure corner: knowing that at midnight (about which time, I forsooke my company) I left the place [which was] ... a verie *Sympathie* of an imagined Paradise. And in the space of one slumbering stéepe, to be left like a desart wildernesse, without any creature, saue sundry Sauage Beastes, portrayed in the Tapistrie hangings, imprest such a heauy passion in my minde, as for the time, I fared as one whose sences had forgot howe to doe their bounden offices: In the ende, to recomfort my throbbing heart, I tooke my Citterne, and to a solemne Note, sung this following Sonet ...

Farewell, bright Golde, thou glory of the worlde,

Faire is thy show, but some thou mak'st the soule:

Farewell, prowde Mynde, in thousand Fancies twirld:

Thy pompe, is lyke the Stone, that still doth rowle.

Farewell, sweete Loue, thou wish of worldly ioy,

Thy wanton Cuppes, are spiste with mortal sin:

Farewell, dyre Hate, thou doost thy selfe annoy;

Therefore my hart, no place to harbour in.

Enuy, farewel, to all the world a foe,

Lyke DENNIS BVLL, a torture to thy selfe:

Disdayne, farewell, though hye thy thoughts doe flow,

Death comes, and throwes, thy Sterne vpon a shelfe.

Flatterie, farewell, thy Fortune dooth not last.

Thy smoothest tales, concludeth with thy shame:

Suspect, farewell, thy thoughts, thy intrayles wast, (blame

And fear'st to wounde, the wight thou faine woul'dst

Sclaunder, farewell, which pryest with LYNX his eyes,

And canst not see, thy spots, when all are done:

Care, Care, farewell, which lyke the Cockatrice:

Doest make the Graue, that al men fame would shun.

And farewell world, since naught in thee I finde

But vanytie, my soule in Hell to drowne:

And welcombe Phylosophy, who the mynde

Doest with content, and heauenly knowledge crowne.

During the time that my thoughtes swounded [swooned][2] with the charme of my passionate Musick: The Sunne decked in his most gorgious Rayes, gaue *a bon Giorno* to the whole troupe: and so many as were within the sound of my instrument, were drawen with no lesse vertue, then the Stéele vnto the Adamant . . . I was driuen foorth of my muse, with a starkeling admyration, not vnlike vnto him, that sléeping ouer a dying brand, is hastelie wakened with the lightening of a thousand sparckles . . . the Poets fayned not without reason, that *Amphions Harp gaue sence vnto stone Walles*. (Whetstone 1582, sig. Hiir–Hiiir)

Whetstone's sonnet overtly contrasts the ephemerality of sensuous social activity with the fixity of the surrounding walls and furnishings. It expresses a sense of wistful melancholy at the evanescence of sensory pleasure, which might be considered a kind of *memento mori* on the fleeting nature of the material world (an idea to which I shall return). While such fictional accounts, like paintings, require some caution (Elias 1989), Whetstone's words hint at a particular relationship between sociability, musical activity, and the interior spaces in which those activities took place. Recreations such as music making or masquing are experiences that shake the senses, echo loudly in memory, and are also profoundly social—the exchange, symbiosis, and dissonance of meanings created and emotions generated through the co-experience of multiple senses in conjunction or proximity. Moreover, the relationship between activity, sensing, and passionate feeling was not straightforward. Recounting the previous night's activities stirs in Whetstone a feeling so powerful, that he was left as 'one whose sences had forgot how to doe their bounden offices'. Yet the only cure for this passion-induced senselessness is the hair of the dog.[3]

As Whetstone demonstrates, wordplay between 'sense' and 'making sense' was not lost on early modern readers. As Mary Carruthers says, '[m]aking sense of something is not the same thing as defining it only in words, for English 'sense' . . . is used for the whole human complex of thought, feeling, and perception, that kind of knowledge which is based in sensory experience' (Carruthers 2013, p. 136). Moreover, the experience of each sense is rarely truly discrete, as they are more often encountered together (Bank 2021, pp. 80–101; Austern 2020, pp. 193–216). My work builds upon the writing of Simon Smith, Eleanor Chan, and others who demonstrate how early modern musicking was a far more visual experience than we often assume (Chan 2023; Smith 2015, pp. 166–84), as well as scholars working more broadly in sound studies and the history of sensing and emotion (Summers 1987; Harvey 1975; Classen 1993). Though hierarchies of the senses were a common topic of discussion in plays, treatises, music, and poetry, I suspect that by scrutinizing sensing events in isolation (music with the ear, art with the eyes, etc.), we are missing something significant, particularly in early modern England. One mode for

accessing such intersensory relationships is by more holistically exploring the spaces in which sensing activities took place.

Tara Hamling has shown how the transmission of imagery from Protestant devotional manuals to the walls and ceilings of houses helped to direct Protestant piety within domestic space (Hamling 2010b, p. 71). Such recurring texts and iconographies not only displayed the piety of the household, but also helped shape it (Hamling 2014, pp. 238–9). As scholars such as De Certeau (1984) and Huizinga (1998) have articulated in their own ways, space is a practiced place. Or perhaps in Whetstone's words, musical activity gives 'sence unto stone walles'. This article explores one avenue for investigating the complex relationships between sensing, sociability, activity, and space through an in-depth examination of a late sixteenth- and seventeenth-century English interior design trend: personifications of the Five Senses. It considers how the Five Senses engaged early modern English subjects in a dialectic between sensory/bodily absence and presence as a mode for exploring the precarious pleasure of holding the passions on the edge of balance. Five Sense iconography is therefore 'incomplete' without the human activity that connects it to one's sensing body.

This study imagines this type of decoration and song within its embodied artworld to understand better what depictions of the Five Senses did in practice for those who viewed and lived with it. As Alfred Gell says, 'decoration is intrinsically functional, or else its presence would be inexplicable' (Gell 1998, p. 74).[4] Whetstone reminds us that spaces such as Great Chambers had not only practical uses, but were also arenas for imagination and play, whether for masques, poetry, games, music, or dance.[5] This article engages with the spatial and ritual framings of Five Senses decoration and musicking, taking into account the feeling, sensing bodies experiencing musical–visual sensory interplay in early modern elite households. I explore further the musical–visual relationship that is referred to by Whetstone: how did sociable, aesthetic, sensing activity 'give sence' to spaces? Yet the relationship between the spatial/visual and embodied activity was not unidirectional. Though it is the music (Amphion's harp) that gave sense unto the stone walls in Whetstone's imagining above, it is the return to the physical space that triggers powerful memories of the past night's convivial activity, persuasive emotions felt so viscerally that they overwhelm his current state of being, triggering the impetus to pull out his cittern once more. The lure of the music, then, draws together the party anew in a continuing process.

Though Whetstone's speaker was playing alone, his action still situates this type of musicking within a social setting, both of memory and in time, that brings people together. While a slightly different context, the relationship between space, collective music reading, and sociability has been explored by Richard Wistreich, who reasons,

> the act of reading musical notation (and, in the case of songs, its associated words) 'back into sound' is, by comparison with most other literary texts, almost always physiologically quite spectacularly dynamic and also usually to some degree both collaborative and communal ... As such, it admirably fulfils Roger Chartier's dictum that "Reading is not uniquely an abstract operation of the intellect: it brings the body into play, it is inscribed in a space and a relationship with oneself and with others". (Wistreich 2012, p. 3; see also Wistreich 2011; Van Orden 2015)

Extrapolating from Chartier, Wistreich argues that music reading (a) brings the body into play, and (b) is to some extent communal, assertions vital to this examination of aesthetic contemplation of the Five Senses.[6] These ideas are central to how music and visual culture worked together to both shape and reflect emotional experiences of recreational, social music making.[7] Part of this is developing a more culturally contingent understanding of the quality of the relationship between materiality, iconography or ideas, and musical experience or the activity of musicking.[8]

## 1. The Five Senses

Visual representations of the Five Senses as an explicit feature of interior design is largely an English manifestation with medieval origins (Nordenfalk 1985, p. 1).[9] Existing

scholarship on the Five Senses in decoration mainly explains its presence as a part of the upper-class English trend of decorating homes in designs inspired by numerical sequences (the continents, muses, humours, temperaments, seasons, virtues, etc.) (Jones 2010, p. 34). Of these, the Five Senses was the most common numerical sequence, with designs usually inspired by continental prints (Jones 2010, p. 34). With a few exceptions in the literature, rarely is the presence of such iconography contextualised within its relevant domestic activity.[10] The Five Senses is sometimes oversimplified as either a Protestant 'warning' against, or conversely a 'celebration' of sensory activity (Page and Fleming 2021, p. 14). While there is plenty of contemporary literature to support caution when sensing, there is something more to unpack when these ideas are contemplated through ornate decorative imagery or evocative music. Such an aesthetic presentation of philosophy on sensing is more complex than a straightforward 'warning'.[11]

　　Writers such as Richard Braithwaite were well aware of the opposing ways a person's behaviour or moral judgement may be pulled when tempted by passionate sensing of the wrong kind. His print *Essaie upon the five senses* (1620) was 'An aduertisement to the devout Reader, vpon the vse of the fiue SENSES' intended to direct sensing towards the heavens (Braithwaite 1620, sig.A5v). He viewed the senses as windows to the soul, 'organs of weale or woe, happy if rightly tempered, sinister, if without limit' (Braithwaite 1620, sig.A3r; Karim-Cooper 2015, p. 217). The precarious and debated balance between sensing and reason leads me to wonder why early modern English people chose to decorate their homes and sing evocative music about this very tension. Classical debates about the relationship between feeling, knowledge, art, and philosophy abounded in early modern writing, art, and music (James 1997; Gouk 1997; Smith 2008; Milner 2011; Sanger and Walker 2012). In the aesthetic contemplations of sensing discussed here, the tension exists not only between godly and sinful contemplation, but also within the very mode of delivery. In *Essaies vpon the fiue senses*, Braithwaite writes about each sense on its own, whereas contemplation of sensing via plays, song, or ornate decoration brings the whole body into play in dynamic and social ways, imposing precariousness through experience and being, rather than simply through reason. This points to a limitation within even contemporary abstractions such as Braithwaite's, lending support to the potential impact of more dynamic traces of sensate existence.

　　In contrast with continental writers such as Marsilio Ficino, Louise Vinge observes that for Elizabethans, 'sensuality was sometimes more important than philosophy, and so they could let "the lower senses" [smell, taste, touch] as well experience beauty and cooperate in the perfection of love in their poetry' (Vinge 1975, p. 72). One may find appeals to these lower senses in song, as well.[12] While the ideal place of sensuality in daily life probably varied within English culture more broadly, for example along divisions of confession and class, there is also a subjectivity inherent to this type of contemplation. What the presence of this more aesthetic, social, and dynamic contemplation of sensing brings is not a single answer for how sensing should be, but a sensation of pleasure within the contemplation itself. As Carruthers asserts, it is 'often the case with pre-modern aesthetic concepts that they embody not a single feeling but are situated between poles of opposite feelings, in a precarious balance easily upset, but one that also provides the energy and force of the sensation as experienced' (Carruthers 2013, p. 60).[13] The precariousness of balanced sensing is not a feeling to be avoided, but one to be embraced. The ability 'to hold that balance is a genuine pleasure, power exercised between delight and fear' (Carruthers 2013, p. 60). Though Carruthers is discussing medieval aesthetics, it is within this binary and the desire for balance that I believe we need to approach aesthetic presentations of sensing in the late sixteenth and early seventeenth centuries as well.

　　*Memento mori*, for example, had a similar function. As Stephen Perkinson has shown, *memento mori* could not only act as a classicising symbol, but also provide anatomical knowledge, reveal traces of dark humour, and allow artists to demonstrate artistic virtuosity (Perkinson 2017, pp. 67, 74). As with the Five Senses, emotional reactions towards *memento mori* were mixed. While they served to remind people of the inevitability of death (and

therefore to live a life towards salvation) *memento mori* also had associations with 'vanity, ephemerality, time, and lust; the impulse towards sermonizing and virtuosic display' (Perkinson 2017, p. 74).[14] Both of these themes are hinted at in Whetstone's sonnet above through his laments on vanities and the persistent 'farewells'. Perkinson explains that 'these multiple and at times contradictory ways of thinking about, responding to, and grappling with death were always there, lying just beneath the surface of *memento mori* images' (Perkinson 2017, p. 74). As with aesthetic contemplations of the Five Senses, they presented no simple or fixed meaning. *Memento mori* allowed subjective contemplation of the complexities of life through death, experienced as a kind of pleasure through the precarious maintenance of balance.

The more nuanced understanding of pre-modern aesthetic experience explored by Carruthers is rarely considered when looking at the seemingly mundane objects of recreation and decoration. Here, Clare Gapper demonstrates the more typical explanation for this type of iconography: '[p]ersonifications of the Elements, the Seasons and the Senses would all have been glossed as aspects of God's creation which provided suitable subjects for contemplation' (Gapper 1998, 'personifications'). There's nothing wrong with this observation, but such lines of questioning often end there. It is unusual that this interpretation has passed as 'enough' for so long given the volume and variety of the Five Senses trope in not only interior decoration, but also verse and song. My research suggests a more complex uptake of meaning than hitherto analysed in any depth.

I am interested to know how this contemplation of sensing felt and looked, and what activities or rituals occurred as a part of it. It seems logical that a more robust look at song and image within its artworld might better contextualise its experience in the home. As will be explained in greater depth later, I suspect that Five Senses imagery invoked an experience akin to a mindful practice—a trigger that provoked an inward observance of one's own faculties in that very moment—a moment to assess the balance of one's sense-ful life—neither warning nor celebration.[15] The artistic representation is, in itself, a manifestation of Carruthers' 'precarious balance' between poles of delight and fear that produce their own kind of pleasure.

*　*　*

References to the Five Senses abounded within stately homes, in both fixed and movable decoration.[16] In the Pillar Parlour at Bolsover Castle, wall paintings of the Senses based upon Flemish prints surround guests in an intimate room for entertaining (Jones 2010, p. 34). An imposing overmantel in the Drawing Room at Langleys, Essex (c.1621–28), depicts Tobias and the Fish, surrounded by the Five Senses (Wells-Cole 1997, p. 167). Ceiling plasterwork in the Long Gallery at Blickling Hall (c.1625) covers a variety of images and ideas inspired by emblem books, including prominent depictions of each of the Five Senses (Wells-Cole 1997, pp. 164–5).[17] Five Sense plasterwork on the ceiling of the State Drawing Room at Boston Manor (c.1623) may share source material with the examples at Kew Palace (c.1630) (Gapper 1998). Levens Hall, Westmorland hosts a large wood carved overmantel including a number of sequences, including the Senses, with a similar diversity of subject matter to wood-carved bannisters of the Five Senses once in Slaugham Place, now in Lewes Town Hall (Wells-Cole 1997, p. 200; Beard 1975, plate 16). These carved bannisters title each figure with their sense using remarkably active versions of the nouns: Seeing, Hearing, Smelling, Tasting, and Feeling.[18] While this list is not exhaustive, it gives an idea of how prolific such iconography was.

The fixity of plasterwork and painted walls can be interpreted as an indication of a more specialised use of interior space, though early modern use of such spaces was often multifunctional (Hamling 2010a, p. 85).[19] Occasionally, fixed decor is useful in determining provenance and therefore by whom images were viewed or made and in what spaces. However, there is also Five Sense imagery in soft or movable decoration. A stunning set of Five Sense tapestries has been at Haddon Hall since the mid-seventeenth century (they are one of two sets of Five Sense tapestry created c. 1630 for Charles I. This set was sold to the Manners of Haddon c. 1649) (Mulherron and Wyld 2012). An embroidered

cabinet featuring Orpheus and the Five Senses (among other numerical sequences) is now on display at the Fitzwilliam Museum (Cabinet c. 1650, Fitzwilliam Museum T.8—1945). There are also solo senses that may have once been part of a set, such as a rectangular panel of Elizabethan petit-point needlework of *Auditus* at Hardwick Hall, that may have once been a cushion (Anonymous Auditus Panel, Elizabethan, NT image reference 385061). Others are incomplete on purpose. Two seventeenth-century embroidered, framed mirrors depicting four of the Five Senses can be seen at Cotehele, Cornwall and Melford Hall, Suffolk (Figure 1).[20]

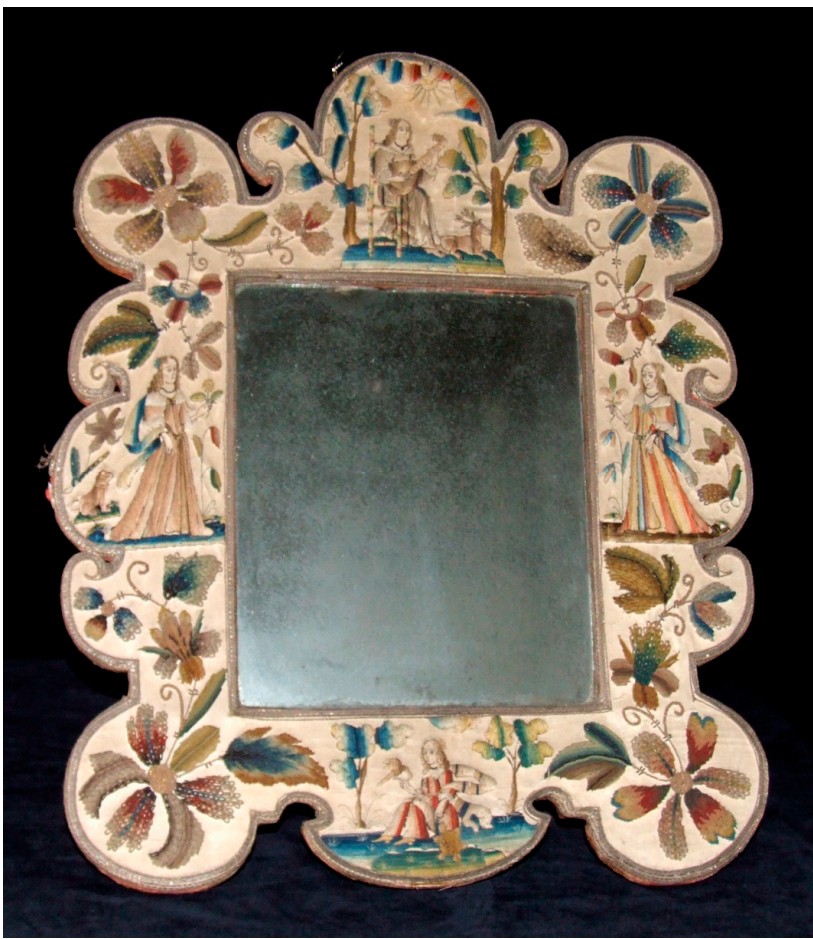

**Figure 1.** Embroidered mirror frame, Melford Hall, 17th century © National Trust/Sue James.

In each of these, *Visus* is missing, only represented by the viewer when she looks into the mirror.

While the Five Senses are represented through static visual culture, their function or fuller realisation requires, even insists upon, an attention to your other senses. It is for this reason I think humanoid personifications of Sensing (or, as they are called at Levens Hall, 'Five Senses . . . Portraited' (Wells-Cole 1997, p. 200)) nearly always accompany other iconographical representations such as animals—an animating human body is crucial (Nordenfalk 1985, p. 4).[21]

## 2. Mind the Gap

One hint for how to approach Sensing iconography of this period is in the two aforementioned embroidered mirror frames at Melford Hall and Cotehele. In these examples, visual imagery is deliberately lacunose, leaving gaps that must be filled by the viewer. Embroidered personifications of hearing, smelling, touching, and tasting surround the mirror. The missing *Visus*, produced by the user's own reflection, takes the place of Sight's

personification. However, this is more than just a charming trick. The role of the viewer, and their position in the manifestation of meaning, is a well-established part of the analysis of art and visual culture. One often finds space within the pictorial plane inviting the viewer to turn attention to their own body, movement, and sensing, as in Caravaggio's well-known *The Supper at Emmaus* (Figure 2), which provokes the urge for one to step up to their places at table, and not let the teetering basket fall. While the embroidered mirrors do not contain the same sense of urgency, they still direct or draw attention inward towards an awareness of what your senses and body are doing in that moment. It might even raise a consciousness of staring at one's own reflection for too long, and therefore the feeling of teetering balance between the potential enjoyment of tending to one's appearance and the temptations of vanity (expressed through a practical and yet beautifully decorated object).

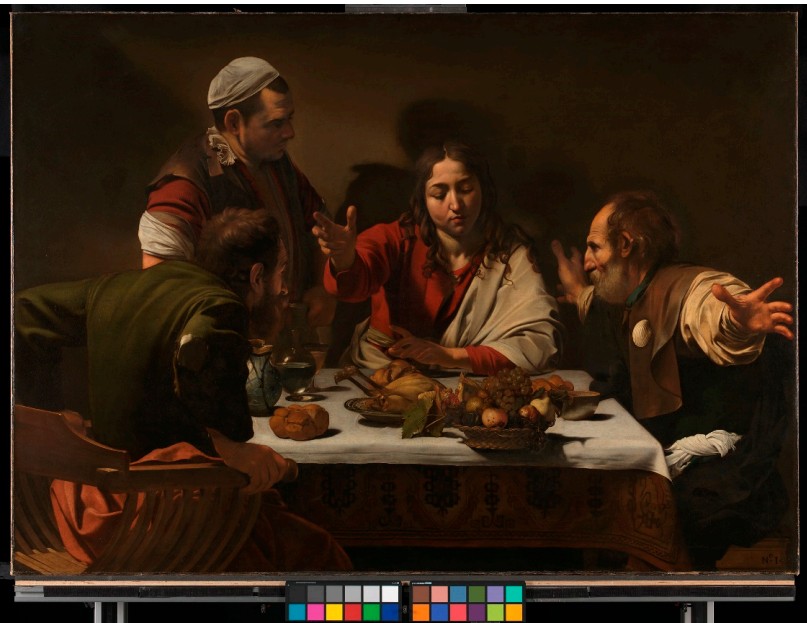

**Figure 2.** Michelangelo Merisi da Caravaggio, *The Supper at Emmaus*, 1601 © The National Gallery, London.

There are other instances where a 'gap' may have been left to allow for a real human body to take the place of a personification. A guest of William Cavendish at Bolsover Castle would be in for a captivating, five-sense entertainment experience from the moment they stepped foot on the grounds. It is believed that guests were greeted by Cavendish in a small unconventional ante room just to the left of the Castle's main entrance. The ante room surrounded visitors in painted wainscoted panelling with striking painted imagery above each of the four walls, inspired by continental prints of the four Temperaments. Sanguine, however, is conspicuously missing, replaced by an image of a temple, much like a theatrical backdrop (Stevens 2017).[22] As Timothy Raylor convincingly argued, this is probably where William Cavendish first presented himself to his guests, a man of Sanguine character, lute in hand, perhaps performing a song, as depicted in the source print (Raylor 1999). With a very specific body in mind, that of Cavendish, this is another instance of iconography playing with bodily presence/absence, as well as the impermanence of the musical activity that completes the visual sense. Favourable acoustics from the wainscotting aside, the room lacks the usual signs of musical iconography—no depictions of instruments, notation, Orpheus, or figures singing.[23]

Even if the learned and astute Cavendish guest did not recognise the other Four Humours (Figures 3–7) from their print sources, one might imagine they would be familiar with the sequence and be able to recognise it. I would therefore argue this still qualifies as an instance of musical–visual culture, even without Cavendish and his lute standing in for Sanguine.[24] Later in the day, guests may have dined or heard more music in the

intimate Pillar Parlour, surrounded by wall paintings of the Five Senses (Figure 8). To what extent are these Five Sense images, too, an invitation to 'fill the gap'? There exists an inherent tension in static depictions of the Five Senses that seems to yearn for either increased awareness of one's own bodily sensing in that moment, or to invoke memories about past sensing experiences. Or is there no 'gap' to be filled in the case of the Pillar Parlour because all five figures are in place?

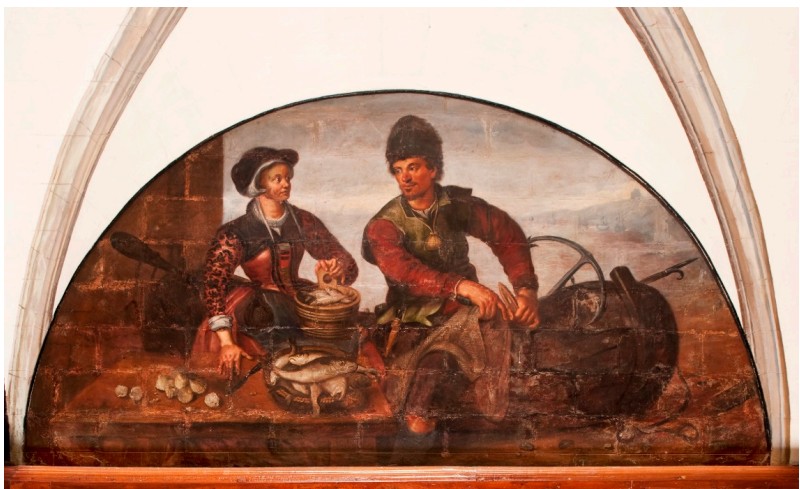

**Figure 3.** Detail view showing the painting illustrating phlegmatic humour in a lunette in the anteroom of the Little Castle, Bolsover. © Historic England Archive. Reuse not permitted.

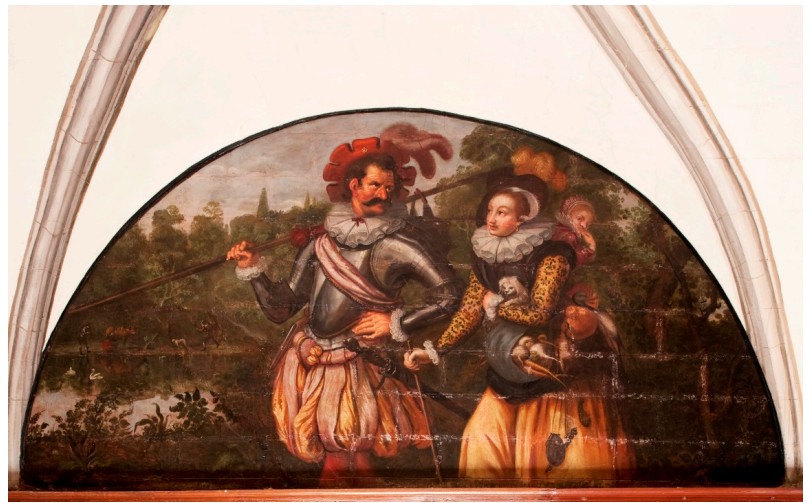

**Figure 4.** Detail view showing the painting illustrating choleric humour in a lunette in the anteroom of the Little Castle, Bolsover. © Historic England Archive. Reuse not permitted.

From the examples of the embroidered Five Senses mirrors and other visual works with similar effect, including the Bolsover Four Humours, I imagine a parallel approach with more 'complete' depictions of the Five Senses might be a fruitful avenue for exploration. I posit, as does Raylor, that such iconography required viewers to pay better attention to their sensing experiences at that moment. Raylor asks if

'[t]he provision of an *actual* banquet (with music, perhaps, to complete the sensory assault), would thus have brought the decorative scheme of the [Pillar Parlour] to life in just the way that the play on the host's temperament had done in the Anteroom'? (Raylor 1999, p. 414)

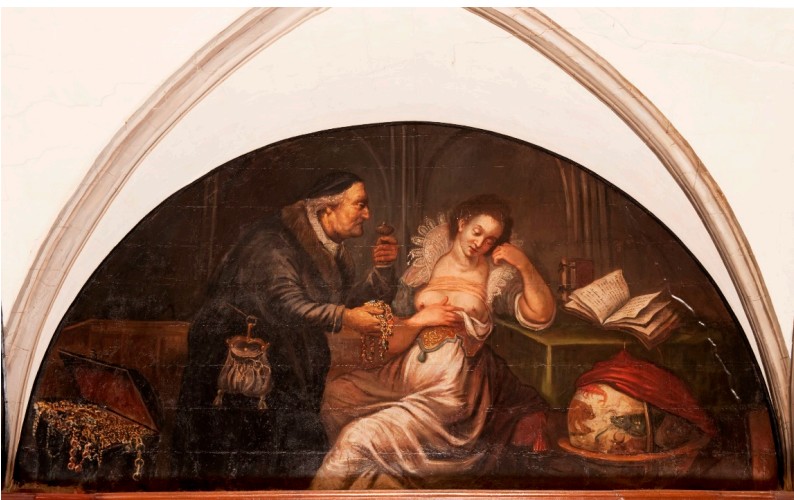

**Figure 5.** Detail view showing the painting illustrating melancholic humour in a lunette in the anteroom of the Little Castle, Bolsover. © Historic England Archive. Reuse not permitted.

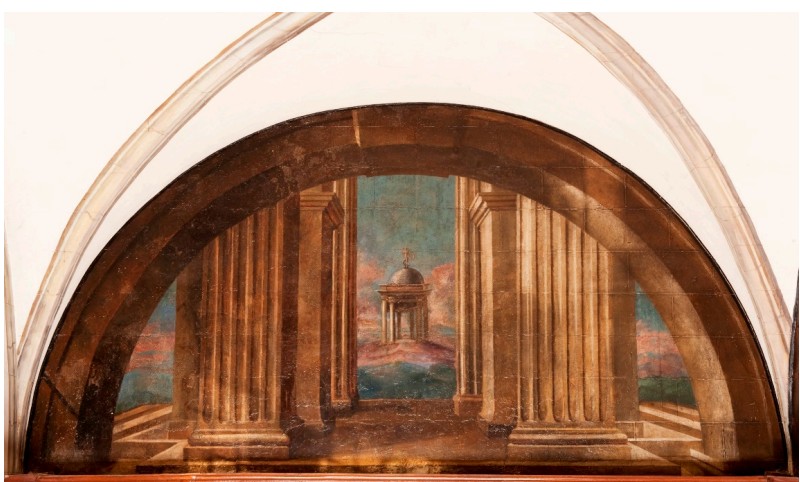

**Figure 6.** The missing 'sanguine'. Detail view showing a painting of columns and a circular building in a lunette in the anteroom of the Little Castle, Bolsover. © Historic England Archive. Reuse not permitted.

What if the song performed had this very space in mind? While there is no extant music, the Song at Banquet that opens Ben Jonson's Bolsover Masque (*Love's Welcome at Bolsover* [1634]), written for performance at the Castle, is

'When were the Senses in such order plac'd?', marked to be 'Sung by two Tenors, and a Base'. (Cutts 1970, p. 53)

CHORUS

If Love be called a lifting of the sense

To knowledge of that pure intelligence,

Wherein the soul hath rest and residence,

FIRST TENOR

When were the senses in such order placed?

SECOND TENOR

The sight, the hearing, smelling, touching, taste,

All at one banquet?

BASS

> Would it ever last!

FIRST TENOR

> We wish the same. Who set it forth thus?

BASS

> Love! (Jonson 1643)

It seems no coincidence that the opening of Jonson's Bolsover Masque asserts the realness of Cavendish's banquet of the senses. As James Knowles observes, this variation on the 'banquet of sense', the treacherously alluring banquet that celebrated the sensory, sensual, and erotic pleasures often associated with Ovid, was regarded as the antitype of Platonism:

> 'Here, Cavendish's 'real banquet to the sense' combines sensory pleasures and a spiritual banquet and so surpasses the normal opposition of soul and senses. The banquet is 'real', that is, material and not simply ideal, stressing the point that his demesne embodies and fulfils Platonic (and, implicitly, royal) ideas'. (Knowles n.d)

Through a mode that itself expires, the song conveys a sense of *memento mori* as the singers lament the impermanence of both banquets and love.

I have been arguing that an aesthetic presentation of the Five Senses, whether in decoration, theatrics, or music, *does* something that purely textual contemplations cannot. Part of this is rooted, as Christopher Small explains, in the ineffable subtleties communicated through the body, face and gesture, as well as vocal intonation, that serve as communicators where words prove insufficient (Small 1998, p. 61). In early modern terms, one might turn to physician and translator Richard Haydock, who in his *A tracte containing the artes of curious painting* (1598) quoted what one might qualify as the 'spirit' of Horace:

> (whence the Poet saith,
>
> If thou in me would'st true compasion breede,
>
> And from mine heavy eies wring flouds of teares:
>
> Then act thine inward griefes by word and deede
>
> *Vnto mine eies, as* well *as to mine eares:)* . . . And, that which is more, will cause the beholder to wonder, when it wondereth . . . to haue a fello-feeling when it is afflicted. (Haydock 1598, sig.r.Aai)

In translating this treatise on painting and art from Giovanni Paolo Lomazzo, Haydock is only paraphrasing Horace in his English verse.[25] It struck me that in Haydock's version, it is the combined powers of ear and eye, words and action (or expression), that best produces wonder and fellow feeling. Both Small and Haydock are, in their own ways, acknowledging the embodied, social dimension of artistic communication. The presentation of the Five Senses as humanoid figures personified taps into this bodily connection, and perhaps also draws attention to the bodies doing those activities within those spaces. In refuting nineteenth-century criticism about the mediocrity of the Bolsover Pillar Parlour Five Senses, Cutts says

> '[t]hese figures are not merely abstract representations of the Senses . . . All five figures are alive in their own right and in surroundings that pulsate with life'. (Cutts 1970, p. 60)

While Cutts may have been referring to the life illustrated within each pictorial plane, the real embodied activity within the space impacts the function of the images with potentially equal power.

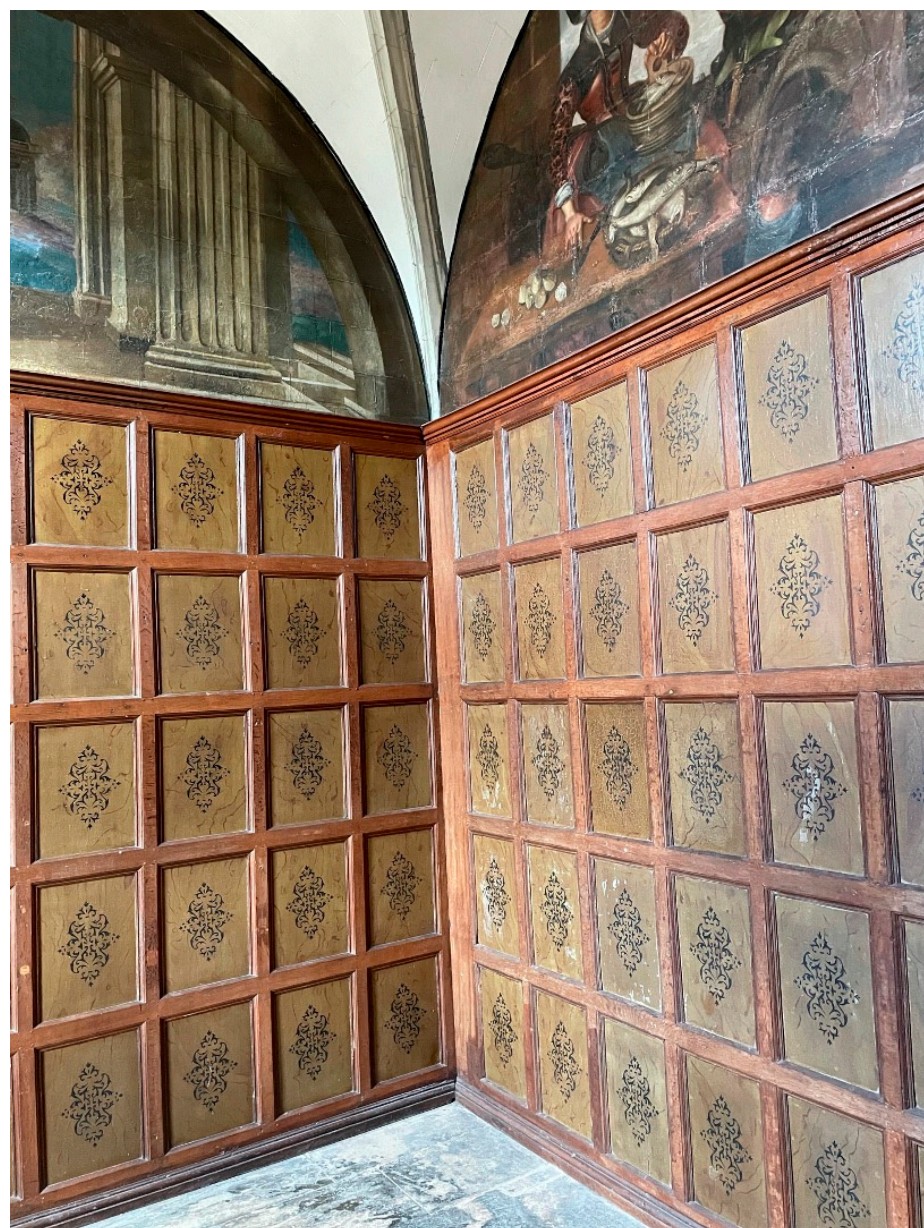

**Figure 7.** Bolsover Anteroom (photo: author's own).

It is not unusual that art asks something of the viewer (i.e., *Emmaus*). Sometimes there are literal gaps that need filling (the Bolsover Four Humours, Five Sense mirrors), and other times the art provokes a tension that seeks resolution in the form of human thought, activity, and a mindful awareness of the viewer's own sensing body. I also suspect that music or other aesthetic presentations of metaphysical questions about sensing have a similar effect. To explore this, I invite a thought experiment. Through this active imagining of musical entertainment alongside the Knole Great Staircase, song and image on the Five Senses within their domestic artworld, I seek to better understand the experience of those who gave life to these social situations.

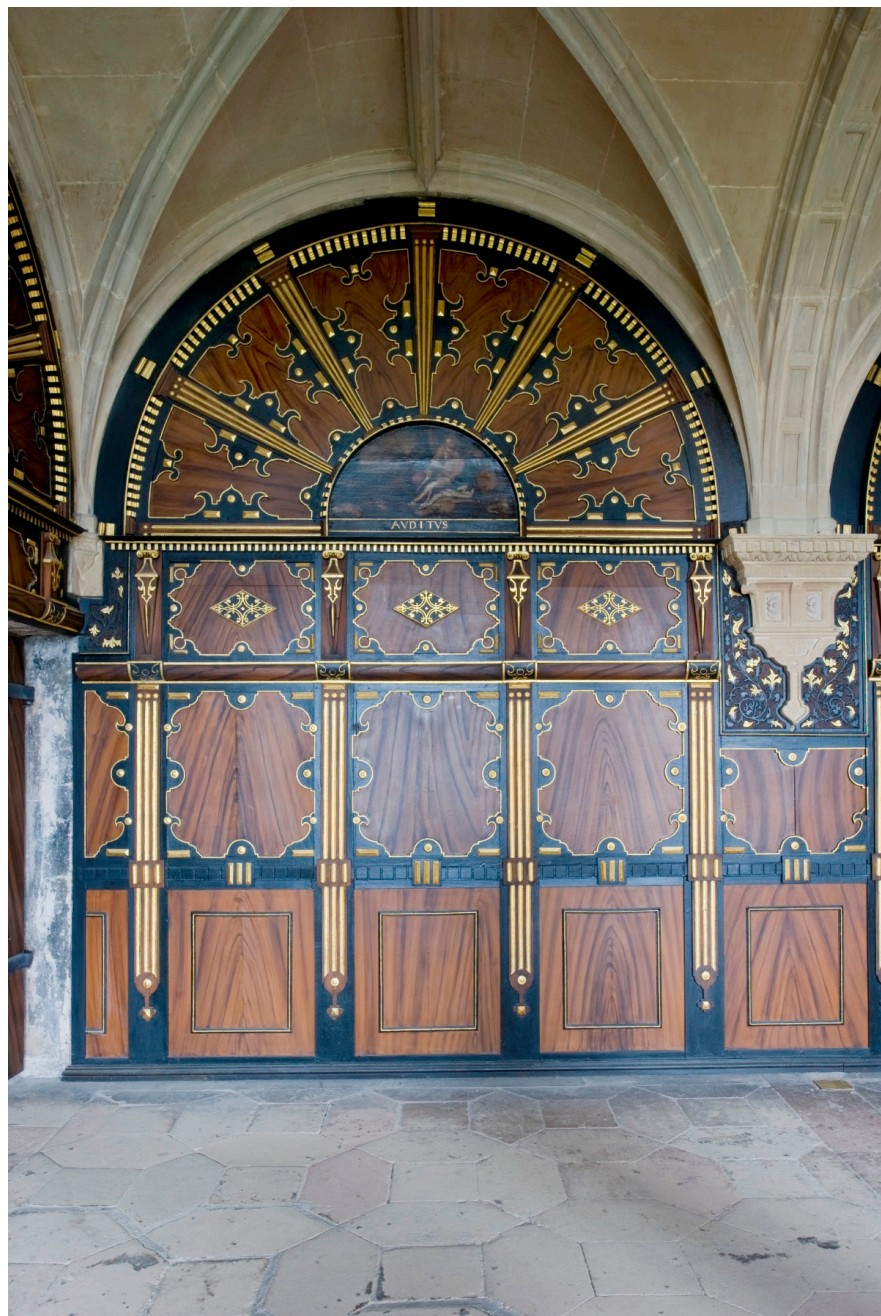

**Figure 8.** Interior view of panelling in the Little Castle's restored Pillar Parlour, Bolsover. © Historic England Archive. Reuse not permitted.

### 3. A Procession of the Senses at Knole House

The following imagining nods to a well-established practice in musicological, organological, and performance-based work that relies on conducting practical experiments to establish how best to perform music using vocal and instrumental techniques approximating those used by early modern musicians. Such opinions about performance practice (often called historically informed performance) are formed not only through material and written evidence, but also through collective and expert listening. The methodology adopted here involves a Baconian kind of empirical approach where knowledge is imparted not only through words but through action.[26] As Braithwaite's report of the senses reveals, early modern English writers have not left overt accounts of such intersensory experience

short of the type described by Whetstone, so these experiences need to be accessed via other methods.

The links between visual culture and music are often elusive and to some extent reliant on subjective experience. Indeed, some scholars deny the possibility of discovering how historical subjects felt emotionally (Rosenwein 2010).[27] Gary Tomlinson suggests the historian's conversation in this area might borrow from anthropology. Both disciplines have the potential to 'widen our understanding of human concerns, beliefs, actions, and so on by juxtaposing our culture with some inkling of another' (Tomlinson 1988). Historical texts, objects, and spaces provide us with that 'inkling'. Without the juxtaposition with our own experience, we cannot hope to interact with, let alone hypothesise about another's experience. As such, my active imagining implicitly draws from my own embodied experiences of musical–social occasions in historic (or neo-historic) spaces.[28]

Though we cannot reconstruct historical experience 'from within', by casting a mould around the object of inquiry with as much context as possible, as Bruce R Smith suggests, perhaps we can get a glimpse of its general shape (Smith 2000, pp. 325–26). For this reason, I would argue that musical-iconographical relationships rarely provide fixed answers or even stable guidance. Yet by placing song and image with shared thematic tropes from the same social sphere in proximity, I hope to glimpse such historical subjectivity.

<div align="center">* * *</div>

Surrounded by a deer park, a version of Knole House in Kent has existed since at least the fifteenth century. After a stint as an archepiscopal then royal palace, the originally medieval estate was eventually purchased in 1605 by the prominent courtier (and second cousin to Queen Elizabeth I), Thomas Sackville, 1st Earl of Dorset. Between 1605 and 1608, Sackville remodelled the interior and exterior in the height of Jacobean style (Parton 2019, pp. 12–13). As Lord Treasurer, Sackville managed the staff at the King's Works and was able to employ some of the same craftspeople to work on Knole (Parton 2019, pp. 8–11). His modifications included ornate alabaster and marble overmantels, moulded ceilings and plasterwork, woodcarvings, and intricate strapwork inspired from continental prints. To connect the medieval Great Hall to the Great Chamber and Long Gallery, he installed a fashionable winding Great Staircase, painted with design work and imagery from floor to ceiling.

The Sackvilles of Knole were a prominent musical family. Thomas Sackville employed many household musicians at Knole and was one of William Byrd's lawyers (Price 2009, p. 18; McCarthy 2013, p. viii). George Abbot noted in 1608 that the Earl 'entertained Musitians the most curious, which any where he could haue' (Abbot 1608, sig.C1v; Hulse 2009, pp. 87–97). Payments to musicians in the Sackville papers include bills for 'strings bought for your lordship's viols and violins' (3 Oct 1607), payments for the musicians' employ over several months, including Christmas, and a payment in April 1608 for '10 of your lordship's musicians for their wages' (Jean 1958, pp. 182–83). When Thomas died in 1608, he made a provision in his will for his musicians: 'And whereas I have and doe entertaine divers Musitions some for the voice and some for the Instrument whom I have founde to be honest in their behaviour and skillful in their profession and who have often given me, after the labors and paynefull travells of the day, much recreation and contentacon [sic] with theire delightfull harmony . . . ' (Jean 1958, p. 183). In this will, Thomas asked his son Robert to employ the musicians and pay each of them an annuity of £20. While it is not a given that Thomas and his family also participated actively in music-making, it remains a viable possibility.

Thomas's grandson, Richard, became 3rd Earl in 1609, inheriting Knole and marrying Lady Anne Clifford in that same year. Musical entertainments continued at Knole under Richard and Anne. Richard paid £10 to Henry Lawes in 1621 'for his half yeres wages due at Mich'as last at which tyme he departed your honours service' (Hulse 2009, p. 96).[29] Moreover, Anne was most likely an accomplished musician in her own right.[30] In 1603, she noted in her memoirs: 'I . . . learned to sing & play on the Bass Viol of Jack Jenkins my Aunt's boy' (Clifford 1992, p. 27). Furthermore, *The Great Tryptic* (Figure 9), commissioned

by Anne later in her life, suggests music was central to her self-image and her conception of the knowledge fit for a young woman of her stature (Hulse 2009, pp. 87–89; Hearn 2009, pp. 2–24). While there is no record of specific pieces or songs played by Anne or her husband, their household and established interest in musical activities such as masquing, give strong evidence that theirs too was also a musical household.

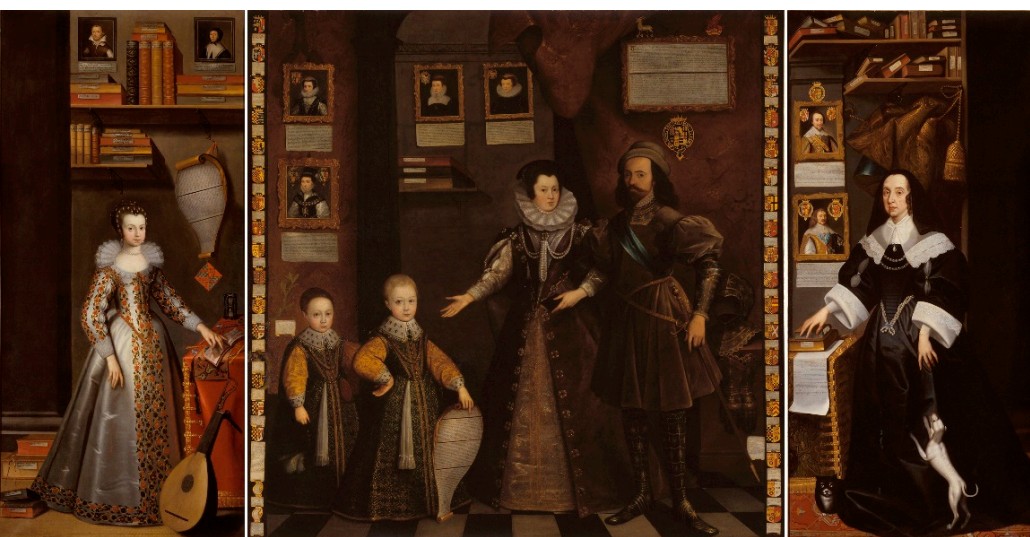

**Figure 9.** Jan van Belcamp Attr. (1610–1653), *The Great Picture* (triptych), reproduced by courtesy of Abbot Hall, Lakeland Arts Trust, England.

\*   \*   \*

We now have the context to proceed to an imagined situation: You are a cherished Sackville guest at their newly remodelled home, Knole, during the period c.1608–13. Sackville's guests are greeted in the Great Hall—perhaps by a galliard played by the Earl's household musicians—wafting down from the musicians' gallery above. When selected guests are called to process to the more high-status receptions rooms upstairs, you are guided up the Great Staircase. In the sixteenth and early seventeenth centuries, the shift in status from the Hall to other types of rooms on the first floor meant the staircase played an important role. The Great Staircase was 'the principal route to the most ceremonial rooms of the house and therefore the focus for considerable display' (Cole 2010, pp. 124, 163; Hamling 2010a, pp. 141–47). Stylish *grisaille* wall paintings of the Five Senses personified as women surround guests as they ascend the stairs. Each personification is reinforced by an animal representative of each sense and in composition with another figure, probably a Greek goddess. At the top of the staircase is a landing with six of the Seven Virtues, Temperance notably absent (perhaps an opportunity for another 'gap' trick?). The bottom of the stairs might seem quite dark, even in candlelight, so depending on the time of day, visibility increases as you ascend. As you step foot onto the staircase, the Five Senses appear in the following order:

The stairs lift you not only towards higher thinking (and closer to the heavens), but also to music, food, windows with a view, and dancing. On the first flight, you pass Smell and Taste (Figures 10 and 11) on your left. Sight is straight ahead at the first landing where the staircase curves to the right. In other Elizabethan houses, including Blickling and Bolsover, Sight is near a prominent window. This is also the case at Knole. Here, Sight (Figure 12) is sandwiched between two windows—your eyes are dazzled with the light as you try to gaze upon it from certain angles. Depending on the time of day, you might try to look upon Sight and instead focus on the lush green landscape outside or simply be unable to see the painting due to the brightness of the sunlight in your eyes. At night, you might struggle to see the painting in full, depending on candlelight, smoke, and shadow. This draws immediate attention to the sense you're experiencing at that very moment. Such

iconography also draws upon the early modern commonplace of eyes as windows to the soul. At the next landing is Hearing (Figure 13) and then finally Touch (Figure 14). It is worth noting that the order of the senses does not follow the most common hierarchies, where one might find Sight and Hearing in the highest seats.[31] Perhaps music accompanies the party ascending the stairs? Do the stairs creak under the collective weight of the group? What smells waft from person to person or room to room? Does your eyesight adjust from the increase in light ascending the stairs? Are you tempted to touch the Sackville leopard as you turn at the landing? (Figure 15).

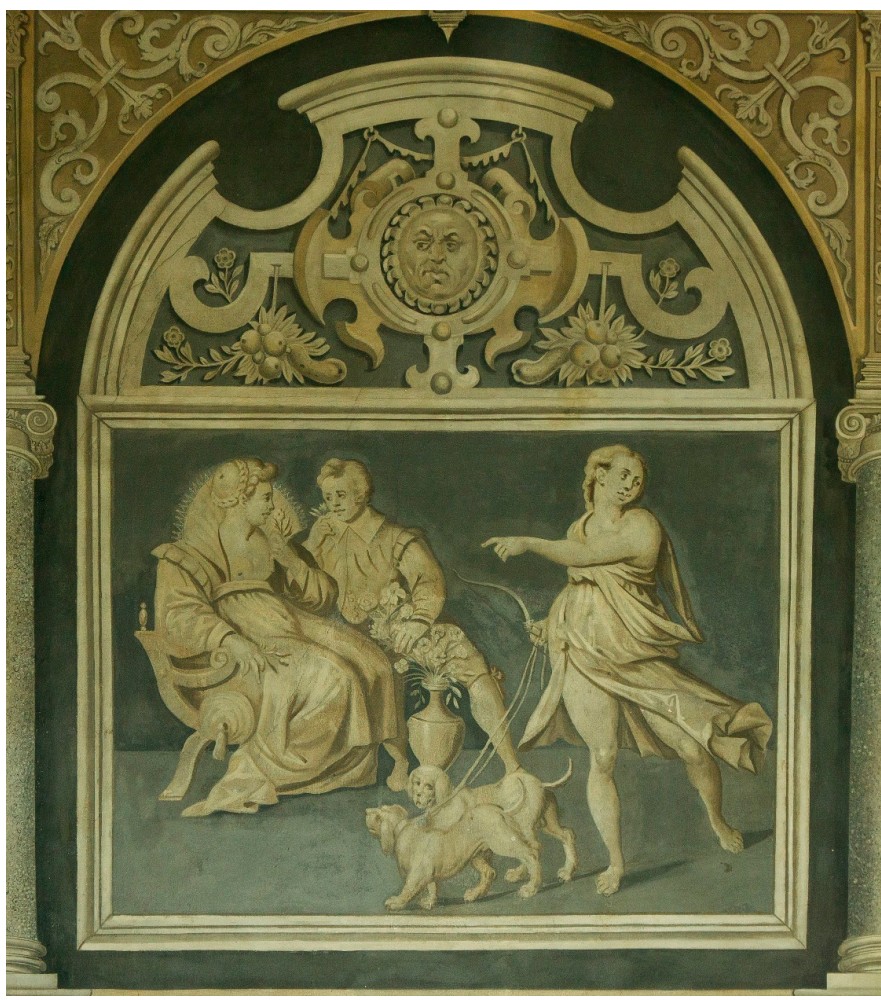

**Figure 10.** Smell, The Great Staircase, Knole. Reproduced by kind permission of the National Trust.

Apparently, it is possible for musicians to travel ahead of the party from the gallery in the Great Hall, through the back corridors and kitchens, to a hidden door at the edge of the landing at the top of the stairs (Figure 16). This landing leads directly to the Great Chamber and Long Gallery, the two main spaces for more intimate entertaining.[32] While this is just conjecture, the spacious landing signals to me a potential musical space. There is plenty of room for a small band to play out of the way whilst guests process into the Gallery or Great Chamber. Hamling has demonstrated how staircases acted as a space for household worship, so it seems a small extension to suggest they also could host other activities (Hamling 2010a, p. 144). Whether artistic licence or actual practice, the masque scene in the Portrait of Sir Henry Unton (Figure 17) shows performance activity involving stairs, so these stairs at Knole could well function as a musical space, and not just as the corridor between more obvious musical spaces.

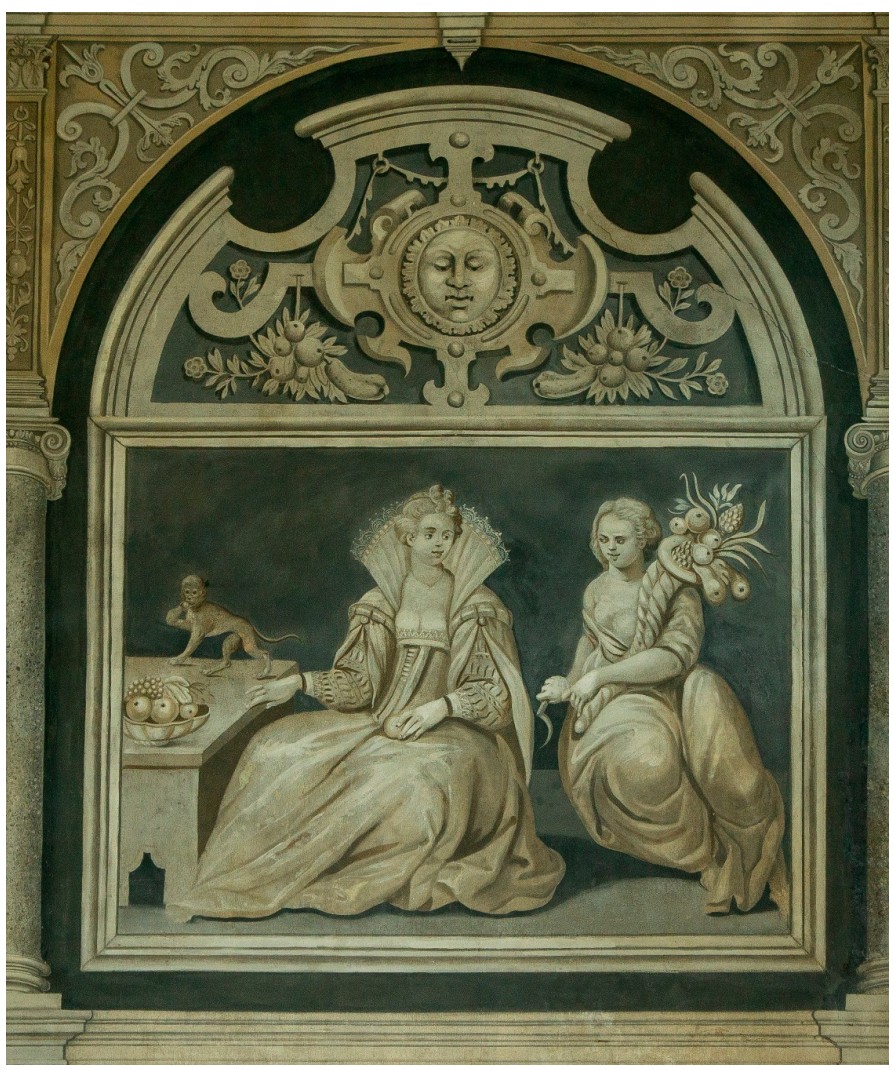

**Figure 11.** Taste, The Great Staircase, Knole. Reproduced by kind permission of the National Trust.

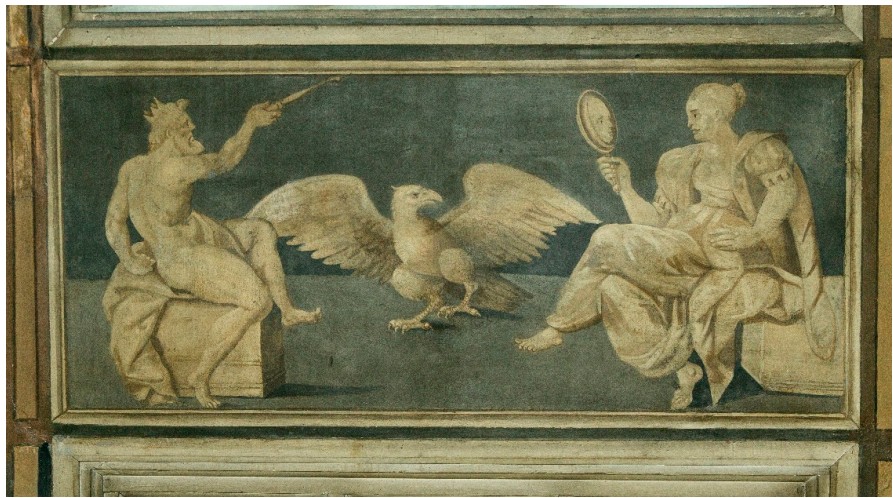

**Figure 12.** Sight, The Great Staircase, Knole. Reproduced by kind permission of the National Trust.

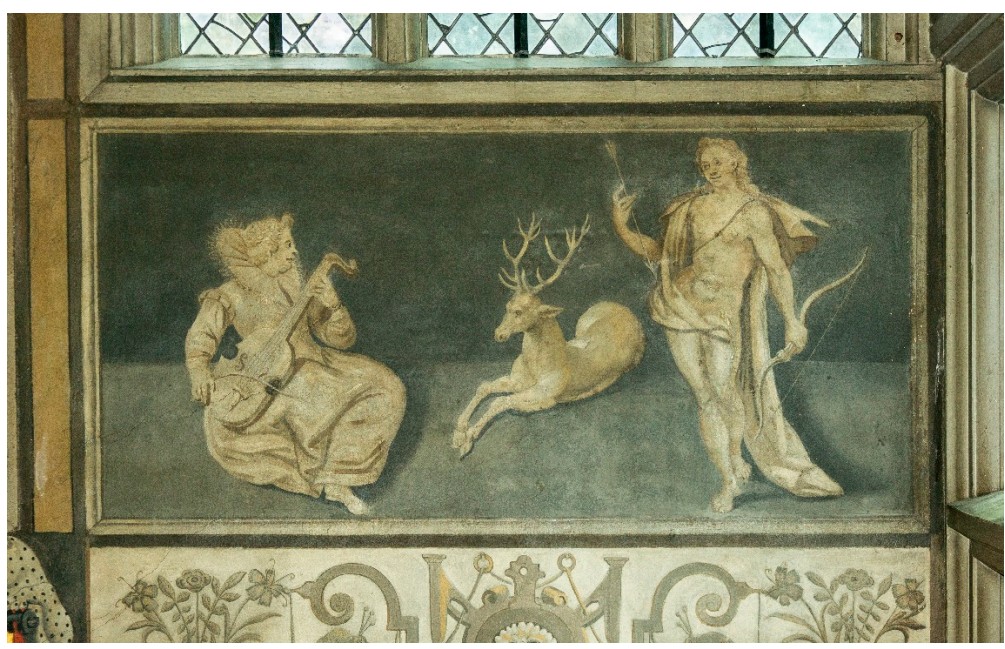

**Figure 13.** Hearing, The Great Staircase, Knole. Reproduced by kind permission of the National Trust.

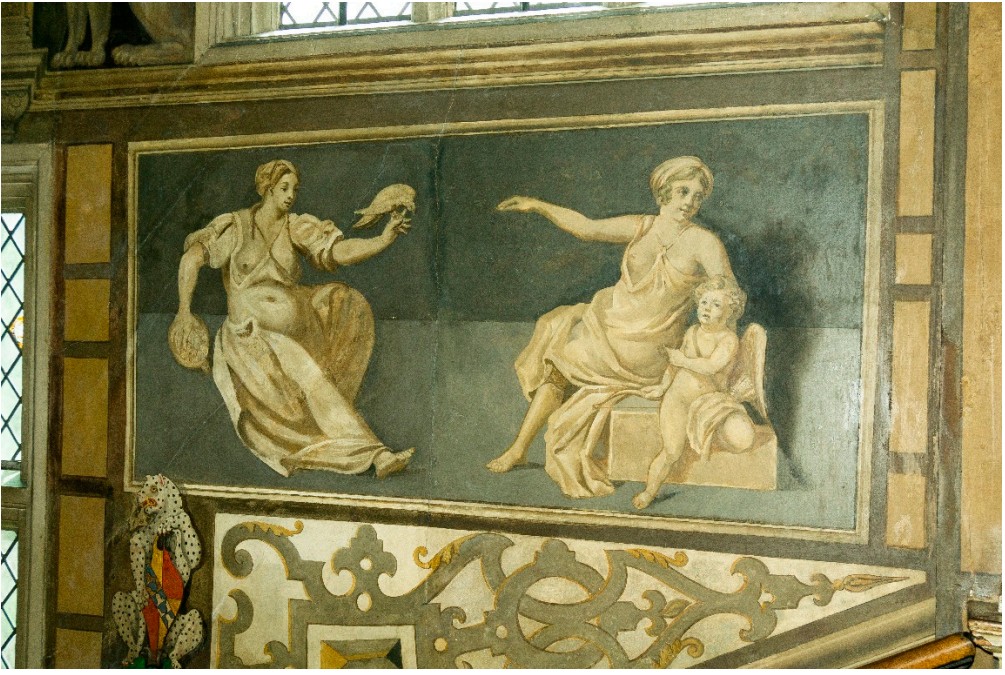

**Figure 14.** Touch, The Grand Staircase, Knole. Reproduced by kind permission of the National Trust.

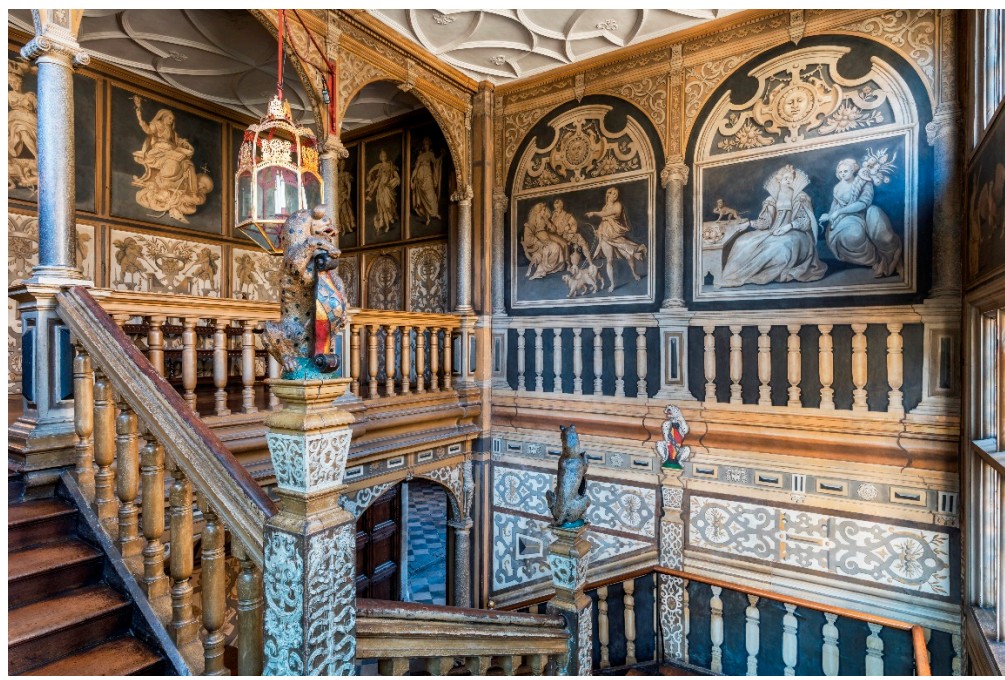

**Figure 15.** The Great Staircase at Knole ©National Trust Images/Andreas von Einsiedel.

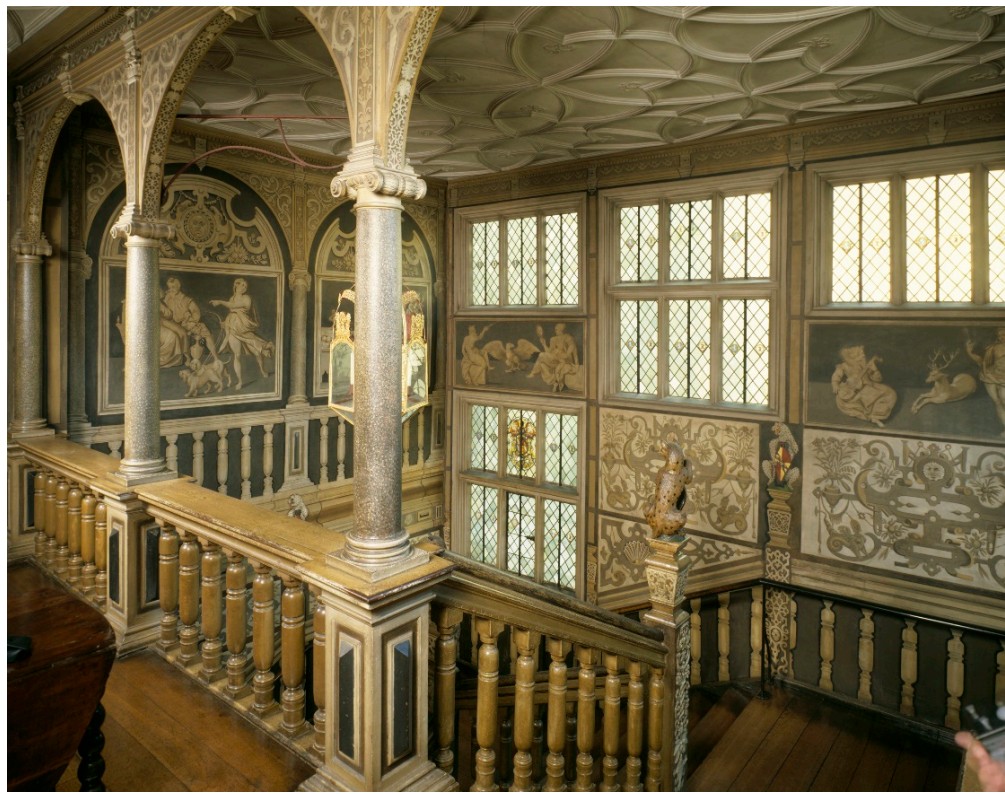

**Figure 16.** The Great Staircase at Knole from the landing ©National Trust Images/Horst Kolo.

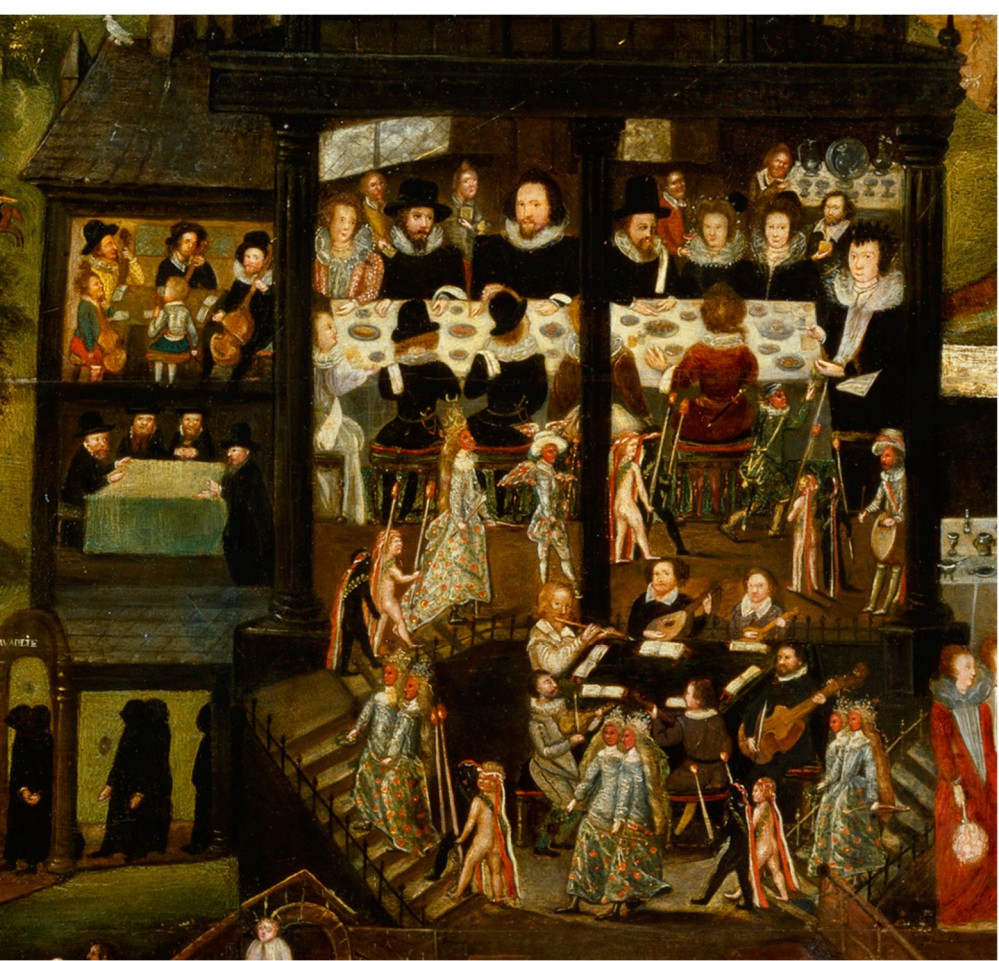

**Figure 17.** Detail of the Portrait of Sir Henry Unton/NPG 710 © National Portrait Gallery, London.

What transpires next in the Great Chamber or even the Long Gallery might be an evening social scenario much like the one outlined in Claude Hollyband's *The French Schoolemaister* (1573), which was dedicated to a young Robert Sackville, 2nd Earl of Dorset (Thomas's son and father of Richard, 3rd Earl). It is possible some aspects of the script were inspired by Sackville family life, as Hollyband was their tutor.[33] Hollyband's dinner scene in particular attends to all five senses in a very present way. Comment is made upon the daughter's looks, the aroma of pungent garlic, the sharpness of the knives, the taste and temperature of the food (Hollyband 1573, pp. 122–40; Wistreich 2011, pp. 230–46). A fire is started, and musical recreation is offered from partbooks.

Roland, shall we have a song?

Yea Sir: where bee your bookes of

musick? for they bee the best corrected.

They bee in my chest: Katherine take

the key of my closet, you shall find

them in a litle til at the left hand:

behold, therebee faire songes at

foure partes.

Who shall sing with me?

You shall have companie enough:

David shall make the base:

Jhon [sic] the tenor: and James the treble.

Begine: James, take your tune: go to: for what do you tarie?

I have but a rest.

Roland, drinke afore you begine,

you will sing with a better corage

([Hollyband 1573](), pp. 127–28)

As described here and in the excerpt from Whetstone, this kind of social musicking took place in a sensory-rich environment. As Niall Atkinson has argued, part of being a good host is tending to the senses and therefore the comfort of your guests; all the senses were inextricably linked to social relations ([Atkinson 2018](), p. 22).

Now, we imagine the music begins.

**John Dowland: 'Tell me, true Loue'**

Tell me, true Loue, where shall I seeke thy being.

In thoughts or words, in vowes or promise making,

In reasons, lookes, or passions neuer seeing,

In men on earth or womens minds partaking.

Thou canst not dye, and therefore liuing,

tell me where is thy seate, Why doth this age expell thee?

When thoughts are still vnseen, and words disguised;

vowes are not sacred held, nor promise debt:

By passion reasons glory is surprised,

in neither sexe is true loue firmely set.

Thoughts fainde, words false, vowes and promise broken

Made true Loue fly from earth, this is the token.

Mount then my thoughts, here is for thee no dwelling.

since truth and falsehood liue like twins together:

Beleeve not sense, eyes, eares, touch, taste, or smelling,

both Art and Nature's forced: put trust in neyther.

One only shee doth true Loue captive binde

In fairest brest, but in a fairer minde.

O fairest minde, enrich'd with Loues residing.

retaine the best; in hearts let some seede fall,

In stead of weeds Loues fruits may haue abiding,

at Harvest you shall reape encrease of all.

O happy Loue, more happy man that findes thee,

Most happy Saint, that keepes, restores, vnbindes thee.

([Dowland 1612](), sig.F1v)

This piece by John Dowland is just one of many that might have been sung in such a social situation. It is scored for solo voice, lute, bass viol, and cantus, altus, tenor, bassus 'Repetition'. This was a four-voice optional 'choral' part, underlaid with text, in which additional singers might join in on a repeated refrain so more could partake in the music-making.[34] The strophic form allows for expressive performance along the singers' interpretation of the words. As Thomas Campion said in Rosseter's *Book of Aires* (1601), '[a] naked Ayre without guide, or prop, or colour but its owne . . . requires so much the more inuention to make it please', citing the importance of emotive or effective performance, or the 'action', as he calls it, in the creation of musical meaning ([Rosseter 1601](), sig.B1v).

In this anonymously authored text from *Second Booke of Songs or Ayres* (1600) (Figure 18), also printed in *Pilgrim's Solace* (1612), the speaker addresses a personification of True Love, asking 'what is the quality of Love? Where does it exist?'. Dowland's song reflects the emotional dimension of knowledge via true love, an earlier mode of thought that arguably 'made space' for later philosophers such as Descartes and Spinoza to craft a conception of knowledge as feeling, but in more explicit terms. Recreational vernacular song, including madrigals, consort song, lute song and other forms frequently discussed sensing and the mechanics of sensing, often ostensibly through love, whether romantic or of a Neoplatonic Godly sort.[35] Questions about the reality generated through the senses and the quality of love also touched on the ontological.

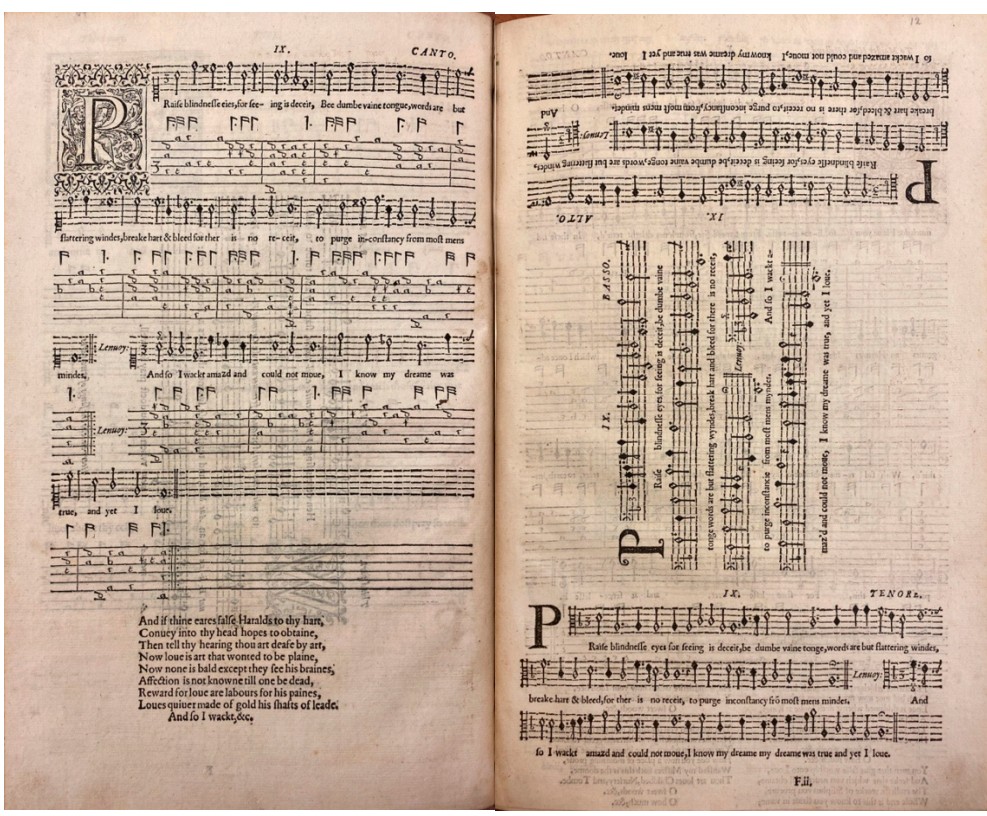

**Figure 18.** John Dowland, *Second Booke of Songs or Ayres* (London: printed by Thomas Este, 1600), sig.Fiv–Fiir. RB 59101, The Huntington Library, San Marino, California. Photo: Ross Duffin.

I do a detailed close reading of this poem elsewhere, so I will not reiterate this here (Bank 2021, pp. 110–13). What I want to focus on, rather than the meaning of the words, is what a performance of such a piece does in a space that visually draws explicit attention to sensing experience. There is something paradoxical about the performance of a song that plainly addresses and even questions sensing, yet is carried by evocative music and within a strikingly decorated space. If you are involved in the musicking, you are singing/hearing about an inevitability—you know falseness and deception abound in matters of art, life, and earthly love. Your senses, including the ones stimulated now—sight, sound, and touch, maybe even taste—are deluding you in this very moment. One is singing cautious words about the knowledge gained through sense perception, but also feeling a more implicit type of knowledge from the heart, as one often experiences through music when it 'striketh' or 'ravesheth' your soul.[36] Through multi-sensory musical experience, we get to practise both types of being and experience delight in maintaining that precarious balance.

Where might the musicians look or gesture on the line: 'Beleeve not sense, eyes, eares, touch, taste, or smelling'? Would musicians or hearers think of the personifications they processed on their way up the stairs? Or would their attention be drawn to the singer's

expression or the sensing in their own bodies: are they making eye contact or reading from a score? Are their finger pads sore from too much lute or viol playing? What scents are lingering from dinner? Is that a friend's perfume? Or is the smell of smoke overpowering? Does a swig of wine help bring more courage for sight reading (as claimed by Hollyband)? Or perhaps they are too fixated on the attractive person across the room to even notice. Dowland's evocative melody appropriately sets the spirit of the text—to what extent do you submit to its beauty even though the words caution 'both Art and Nature's forced: put trust in neyther'? Within the binary set up by Carruthers, this type of aesthetic experience presents the mind with words that describe one pole of feeling, one of caution, and presents the body with its opposite—a sensory experience of beauty in the form of poetic words and evocative music that, to use Francis Bacon's term, 'striketh' the heart/spirits directly. By the time the guests are fully engaged in recreational musicking on the first floor, they have already been introduced to this kind of thinking and feeling through the visual decoration on the same theme. The energy and sensation produced through this precarious tug-o-war produces its own kind of pleasure.

The poignancy of such a tension was surely not lost on those singing or performing this music. This combination of Dowland's song and Knole's Five Senses imagery is just one imagining of many images and songs that may have been experienced within the same social spheres of a learned person of privilege in early seventeenth-century England. Indeed, we even saw echoes of these themes in the Bolsover Masque Song, as well as in Whetstone's sonnet. The specific combination does not matter as much as acknowledging that individuals encountered ideas and reference to figures such as the Five Senses in multimedial ways. This allowed for an understanding of sensing that accrued through a variety of senses and modes and over a period of time. Moreover, the unreliable quality of the senses explored through these songs bring into question the very (im)materiality of the paintings on the walls.

While there is no 'proof' that this song was performed in this house, the individuals that regularly encountered music by well-established composers such as Dowland and who walked the staircase at a residence such as Knole shared social circles, particularly for a well-established, musical family such as the Sackvilles.[37] These were the very types of people that brought life to Dowland's music and whose recreational domestic activity might have brought 'sence' to Knole's stone walls.

## 4. Considering the (Im)Material: Sensing Sight and Sound

Susan Boynton and Diane Reiley explain that in medieval England, both the idea and experience of music were central to the received meanings of many visual works. The 'links between art and music, rather than directly representational, are often indirect and elusive. The possibility of multiple readings complicates the construction of meaning in visual references to music, and the resulting indeterminacy is often difficult to resolve' (Boynton and Reilly 2015, p. 15). It is my observation that in the early modern period, the relationship between music and visual culture remains elusive, yet there are ways to begin to access it. Though difficult to resolve, it should be possible to glimpse, particularly if we move beyond iconographic meaning. Viewing 'music' as 'musicking', to use Small's term, or at least viewing music as an activity rather than just as a text or thing is another starting point. Written texts are of course an important part of triangulating the function of a trope such as the Five Senses in multiple formats. However, as Helkiah Crooke noted in 1616, a shared, animated experience of knowledge, such as in a play or musical performance, promotes a recreative delight as well as learning: 'Wee are more recreated with Hearing than with Reading . . . For we are wonderfully delighted in the hearing of fables and playes acted upon a Stage, much more then if wee learned them out of written books' (Crooke 1616, sig.Ooo2v; see also Austern 2020, p. 171). Part of the power of performed engagement with this type of knowledge comes from its social dimension. As Wistreich has noted, music-making's (particularly partbook singing's) sociable, collaborative production make it 'one of the most sophisticated and complete examples of communal engagement

with written text' (Wistreich 2011, p. 224). Or in Crooke's words: 'there is a kind of society in narration and acting, which is very agreeable to the nature of man, but reading is more solitary' (Crooke 1616, sig.Ooo2v). Another part of the power of performed engagement with knowledge comes from the fact that it provokes responses from multiple senses (as Haydock suggested), leading to 'fellow feeling'. Crooke also observed that a live performer's voice is more affecting 'by reason of his inflexion and insinuation into our Sense' (Crooke 1616, sig.Ooo2v). Therefore, 'those things which be heard, take a deeper impression in our minds, which is made by the appulsion or ariuall of a reall voyce'. (Crooke 1616, sig.Ooo2v; Richards 2019). Crooke, Haydock, and Whetstone all give reason as to why a look at multiple types of sensing or sensory co-experience might give us a fuller and more powerful look at early modern sensory experience. There is something communicated voice-to-voice, face-to-face that goes beyond semantic communication and bonds us together.

Early modern English creatives did consider sensing events together, as shown not only by both Whetstone and Haydock, but by others including composer Richard Alison, who wrote in 1599:

> And that our meditations in the Psalmes may not want their delight, we haue that excelle[n]t gift of God, the Art of Musick to accompany them: that out eyes beholding the words of Dauid, our fingers handling the Instruments of Musicke, our eares delighting in the swetenesse of the melody, and the heart obseruing the harmony of them all: all thse doe ioyne in a heauenly Consort, and God may bee glorified and out selues refreshed therewith. (Alison 1599, sig. aiir–aiiiv)

The more senses that are involved, the more immersive the emotional experience. While Five Sense imagery often included overtly musical iconography, such as a woman playing a lute, it also tapped into wider ideas about sensing, knowledge, and recreational life that went beyond musical symbolism. Music's connection to the Five Senses was not only about hearing, as music often was experienced in sensory-rich environments, whether in the home, a local pub, an outdoor festival, or church.

This experiment suggests the benefit of looking at music and visual culture together. In locations such as Parlours, Galleries, Great Chambers and Drawing Rooms, or even Great Staircases, large-scale detailed plasterwork or wall painting of the Five Senses often appeared in or between sites where musical entertainments took place. Material and visual evidence shows that Galleries, for example, brought together the material pleasures of sight and sound—Unton's Wadley House inventory lists 15 pictures as well as a chest of virginals within that space (Nichols 1841, p. 25). Whetstone has shown us how sociability was engrained in spaces, how aesthetic activity was one of the ways through which people emotionally mediated activity and place, as well as how visual and musical cues can trigger memories and mindful being that result in action (in Whetstone's case, further musicking).

While there is little doubt that plasterwork or wall painting communicate identity and status, there is also more to understand about the relationship between how such decoration was or is experienced alongside the activities that took place in the spaces where it was situated. Musical-visual culture is not only something to be taken seriously as material history, as an indicator of who played what and where, but also as a signifier and trigger for embodied activity. Artistic engagement with the Five Senses asks your attention to shift to your own human sensing and your own body in order to take effect. Without this step, the imagery is flat. In modern terms, these visual cues are almost in effect a mindful practice—this is more than guidance towards the 'contemplation' of external ideas, because representations of the senses, specifically, ask you to think about your own sensing body in that moment.

Sensing iconography and other aesthetic presentations of sensing philosophy such as song present the body with an inherent tension. Sure, they invoked contemplation of sensing, but not as a prescribed 'warning' or 'celebration'—the tension between the space, text, activity, and iconography demonstrates this (in both the Dowland song and the specific space in which it might occur, as we have seen). It is a curious and very

human conundrum to be leaned into, rather than shied away from (as with *memento mori*). Ultimately, one arrives at knowing through a combination of feeling (through the activity) and reason. Objects of visual culture act as a reminder of the felt experiences that took place in those spaces where they are situated, but viewing them is also a sensing activity in itself. These multiple and at times contradictory ways of thinking about, responding to, and grappling with the emotions can perhaps best be enacted and experienced (in a controlled environment) through recreational activity and artistic representation. This nexus including place, space, ideas, language, and context should continue to bolster reconstruction of early modern thinking about self-experience so that modern subjects can interpret past representations of interior life with more nuance.

**Funding:** This research was funded by The Leverhulme Trust grant number ECF-2020-074.

**Data Availability Statement:** Not applicable.

**Conflicts of Interest:** The author declares no conflict of interest.

## Notes

1  Whetstone's print is dedicated to Elizabeth I's Lord Chancellor, Christopher Hatton, who was also the dedicatee for Byrd's 1588 songbook.

2  In Mark Girouard 's *Life in the English Country House* (Girouard 1978, p. 94), Girouard interprets 'swounded' as 'surrounded', which also makes logical sense. Thanks to one of my reviewers for suggesting 'swooned', however. "swound, v.". OED Online. March 2023. Oxford University Press.https://www-oed-com.bham-ezproxy.idm.oclc.org/view/Entry/196070?redirectedFrom=swounded (accessed 24 April 2023).

3  There were also those sceptical of this type of mollification through music, such as Richard Braithwaite, who wrote, 'externally sounding accents, though they allay the passion for an instant, the *note* leaues such an impression, as the succeeding discontent takes away the mirth that was conceiued for the present' (Braithwaite 1620, p. 8).

4  Gell says '[i]n place of symbolic communication, I place all the emphasis on *agency, causation, result*, and *transformation.* I view art as a system of action, intended to change the world rather than encode symbolic propositions about it' (Gell 1998, p. 6).

5  Huizinga discusses how the card-table, tennis court or theatrical stage are all arenas in which special rules apply, 'temporary worlds . . . dedicated to the performance of an act' (Huizinga 1998, p. 10).

6  Wistrich's study focuses on the physiological intensity and the sociability of a *collective* reading of music from part books and is more focused on the readers' collective experience than that of any listeners.

7  Though a distinct topic, Gillian Woods' chapter on visual engagement with dumb shows in early modern theatre explores how active spectatorship allowed for multisensory engagement with performance (Woods 2019).

8  An academically rigorous exploration of what is imaginable or possible when faced with scarce or ephemeral evidence is an area of some controversy. Noemie Ndiaye, adapting her use from Saidiya Hartman, describes this research approach as 'subjunctive' (Ndiaye 2022, p. 25).

9  Wells-Cole surmises these personified numerical sequences became popular in post-Reformation England when saints and martyrs were banned (Wells-Cole 1997, p. 102).

10  Hamling is an example of an exception within the study of early modern English material culture, as she contextualizes domestic plasterwork within Protestant household activity and culture.

11  Contemporary literature that addresses the Five Senses includes: Richard Braithwaite, *Essaies vpon the fiue senses* (London: printed by E.G[riffen], 1620), Thomas Tomkis's play *Lingua: or The combat of the tongue and the five* senses (1607), Edmund Spenser's "House of Alma" in *The Faerie Queene* (1590), John Davies, *Nosce Teipsum* (1599), Phineas Fletcher, *The Purple Island* (1633).

12  In addition to dozens of examples on hearing and sight in English song, touch is pondered in songs by East, Farmer, Campion, Weelkes, Danyel, Dowland, Pilkington, Jones, and more. Taste is considered by Tomkins, Dowland, Rosseter, Peerson, Ford, Hume, Jones, and Bateson, and smell by East and others.

13  She goes on to explain that such underlying precariousness drives the opposing clauses of Augustine's description of sensuous pleasures in Augustine, *Sermones*, 159.2; PL 38.868, on Romans 8:30–1.

14  Perkinson points out that they could be found 'in the gruesome ivories with their leering skeletons, oblivious lovers, and starkly moralizing inscriptions' (Perkinson 2017, p. 74).

15  My use of 'mindful' throughout this article relies on a modern readers' understanding of what it signifies—a present knowledge of one's inner state—and does not imply this was an early modern term.

16  There are also instances of Five Senses decoration in the homes of the upper middling sort. Five Senses wall paintings, originally from Park Farm, Cambridgeshire show female personifications of each sense with a motto below, warning against the dangers

of particular activities for women specifically. For example, Taste shows a well-dressed woman smoking a pipe with a relaxed expression. Below her it says: 'Non sence is non sence, though it please my mind,/and is not proper for this sex and kind'. Hearing says: 'Maydes must bee seene not heard. So am I/I am sure? you doth not heare my melodye'. While it is tempting to interpret these as straightforward Protestant warnings, I pick up on a touch of tongue-in-cheek humour in its presentation, so there is more to unpack here. Note: Malcom Jones cites these as from Hilton Hall, Huntingdonshire.

[17] Gapper says the Blickling Five Senses are from plates illustrating the Five Orders with the Five Senses in *Architectura* by Hans and Paul Vredeman de Vries, 1606–7. They are depicted in P Fuhring & G Luitjen (eds), *Hollstein's Dutch and Flemish Etchings, Engravings and Woodcuts 1450–1700*, Vol XLVIII, Vredeman de Vries, Part II, 1572–1630, Rotterdam, 1997, plates 615–19.

[18] These more active terms are also used in late seventeenth-century prints of personifications in Dutch costume, three of which are at the Wellcome Collection: Seeing, Feeling, and Tasting (Jones 2010, p. 37).

[19] Though as one reviewer rightly points out, the multifunctional character of such spaces in is part what imparts such dynamism to their sensory iconography.

[20] There is a later seventeenth-century embroidered mirror at the Met Museum that might be related to these Senses-embroidered mirrors, though this one has some oddities that are outside the scope of this essay (Morrall and Watt 2008, pp. 216–21).

[21] Nordenfalk observes that there were other tropes (a rose, etc.) that represented the senses in the Middle Ages. Sometimes caricatured body parts were also used in this earlier period, but I still see this as different from personification.

[22] Crosby Stevens has written about Bolsover's decoration and its relation to theatre sets.

[23] As Francis Bacon noted, wainscotting is the superior acoustic material for musical entertainments: 'Musick soundeth better in chambers wainscotted, than hanged' (Bacon 1860, p. 246).

[24] The Bolsover Four Humours provokes several questions to be asked at another time: Are such paintings or spaces musical 'objects'? Can iconographies or bodies become 'musical' simply by accepting a visual, imagined invitation?

[25] Höltgen says Haydock's writing 'created a work of a highly personal character' (Höltgen 1978, p. 19). Lomazzo's original quotes a shorter passage from Horace in a section on decorum and sympathetic affect: *si vis me flere, dolendum estprimum ipsi tibi: tunc tua me infortunia laedent* (If you would have me weep, you must first feel grief yourself: then, will your misfortunes hurt me) (Horace 1926, pp. 458–59).

[26] Thanks to one of my reviewers for this astute observation.

[27] It is worth noting that the word 'emotion' and concept of aesthetics were not terms available to early seventeenth-century subjects, but I rely on their use as signifiers to assist in the process of thinking and writing about historical subjectivities.

[28] I have experienced a version of the pleasure produced through 'precarious balance' in my own musical–social life. I regularly get together with friends to sight-read polyphony (and sometimes madrigals) in both social and liturgical contexts. This has included singing the music of Sheppard where he was employed at Magdalen College Oxford, Mundy and Tallis in St Mary-at-Hill, and Tallis's 'O Lord in Thee is All My Trust' in the presence of the Eglantine Table at Hardwick Hall, or Byrd's Mass for Three Voices in my own home after a dinner party. The perfect cocktail of musical achievement, sociability, and space come together to produce what might be called in psychological terms, 'group flow' (Sawyer 2006, p.157).

[29] KA Sackville Manuscripts U269/A1/6.

[30] Lady Anne Clifford's musical life is explored in greater depth by Lynn Hulse in "In Sweet Musicke Did Your Soule Delight" (Hulse 2009, pp. 87–97). Anne Clifford was also connected to well-known musicians through family. Her uncle Francis was Byrd's patron for his 1611 songbook (Price 2009, p. 220).

[31] While this may have been for the sake of the Sight trick, I doubt their order would have been random.

[32] This was conveyed to me by one of the Knole volunteers, but I have yet to be able to consult a floorplan.

[33] Between scenes, the narrator addresses 'you Master Robert' (Hollyband 1573, p. 142).

[34] Anthony Rooley's The Consort of Musicke recording includes this optional multi-voice refrain. *Dowland: The Collected Words* (Anthony Rooley's The Consort of Musicke 2007).

[35] Other examples of English song that question or contemplate sense perception include: Byrd, "O you that hear this voice" and "Where fancy fond" (1588), Weelkes, "Like Two Proud Armies" (1600), Bateson, "Love is the fire that burns me" (1618), Peerson, "Love is the delight" (1630), Coprario, "Deceitful fancy" (1606), Porter, "Tell me where the beauty lies" (1632), and more. Bank 2021, pp. 106–20.

[36] As Francis Bacon wrote, 'the sense of hearing striketh the spirits more immediately than the other senses' (Bacon 1900, II.114).

[37] For example, Thomas Sackville may have been acquainted with Robert Cecil and/or Robert Sidney, two of Dowland's patrons, through courtly connections including Thomas Bodley, William Cecil, Penelope Rich, or even the monarchs themselves. Price has shown how families of similar stature, such as the Cavendishes, had Dowland's music in their libraries (Price 2009, p. 116).

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
