# Peer review of "“Amphions Harp gaue sence vnto stone Walles”: The Five Senses and Musical–Visual Affect"

_arts, 2023_

Round 1

Reviewer 1 Report

This is an excellent article, and I thoroughly recommend publication in its current form (it just needs some very minor copy-editing to pick up one or two incomplete references and the odd typo/slip). It reads its (multi-sensory) sources persuasively and subtly, and in doing so models some novel approaches to thinking about cultural encounters with sights (both static visual art and the sights of performance). Above all, its reframing of 'five senses' depictions not (only) as representations to be read for meaning, but as invitations to the viewer's own embodied sensory practice (and self-reflection thereon) represents a significant and meaningful step forward in how we understand these artworks and the early modern cultural practices they might have supported. It should have real impact on scholarship across early modern studies, including but not limited to art history, musicology, cultural history, and theatre/performance studies.

The article is thoroughly grounded in relevant scholarship, finding particular purchase in applying some of Richard Wistreich's insights to new contexts. I would perhaps also acknowledge Katie Bank's important work in this area (see ch. 2 of Knowledge Building in Early Modern English Music). I would also encourage you to look at Gillian Woods' chapter on visual engagement with dumb shows in Stage Directions and Shakespearean Theatre. Her topic is distinct from yours, but her model of active spectatorship may potentially be worth mentioning as part of the wider project to think in multisensory ways about early modern performance.

A couple of smaller local points:

p. 2.72 – Does 'swounded' not mean swooned/overcome (see OED, 'swound, v.), rather than 'surrounded', as you gloss in square brackets?

p. 5 – Does the disjuncture you track here between Bra(i)thwaite's sensory caution and the cultural tendency to represent in decorative art and song the very tension with sensate existence in the world that this creates perhaps suggest the limits of Braithwaite's abstractions for getting purchase on early modern cultural experiences, as well as the importance of the kinds of sources you explore in this article?

p. 8 – Note that there are a couple of missing references to fill in on this page; you might also cite Alison directly in the final section, rather than via Smith.

Author Response

Response to Reviewer 1

Many thanks for your time and expertise in reviewing my article.

I have looked at Gillian Wood's chapter, and I can indeed see how her model is of value when thinking about multisensory performance. Thanks for this suggestion.

re: "swounded" - excellent point! Mark Girouard also interpreted this as "surrounded", but I do think you're probably right here.

re: pg 5/Braithwaite - indeed. I've tried to make this relationship (and the importance of my sources) clearer.

re: pg 8 - done.

Reviewer 2 Report

This is a stimulating essay employing well-chosen visual and primary sources, and which fits nicely into the im/materiality theme of this volume. In my estimation, though, it’s not quite ready for publication. As it stands, the essay is disorganized and its writing at times rushed; it will need considerable restructuring and editorial work to bring out its main arguments in a more convincing fashion, while tying them more explicitly to the issue’s theme. As matter of courtesy, the author should not neglect in the future to illustrate the text; this time-strapped reviewer was not terribly charmed to “fill in the gap” by looking up all the missing images. Furthermore, the author must take greater care to properly anonymize the text for the review process (p. 16, l. 583 and note 82).

Continuing onto the substance of the essay:

To start, the author must state more clearly and earlier on the driving argument of the paper as well as its connection to the issue’s theme of im/materiality. The main argument, never quite spelled out, is that a dialectic between sensory/bodily absence and presence, between permanence and impermanence, suffuses the design and decoration of Tudor/Stuart domestic spaces. These are spaces whose multisensory visual iconography allude to and anticipate the musical performances that animate them, while leaving conspicuous visual voids once the party is over. (On a related note, the author might try to come up with a more to-the-point subtitle, something like “Between Sensory Absence and Presence in Spaces of Early Modern Domestic Performance”. Furthermore, the abstract should also be rewritten. At the moment, it wastes too many words on George Whetstone. The abstract also suggests that “active imagining” and five senses imagery at Knole House were the essay’s main subject; this is misleading, since the Knole House discussion makes up no more than 1/3 of the paper and only appears well into the essay.)

A clear thesis statement could potentially come right after the long block quote by Whetstone, which contrasts the brevity and liveliness of an imagined multisensory feast to the fixity of architecture and its furnishings. Whetstone’s verses themselves could be put to better use. They are pervaded by a sense of melancholy at the evanescence of sensory pleasure, a melancholy that persists in what may best be described as a “memento mori” sonnet on the fleeting nature of the material world. (It might make sense for the author’s later discussion of memento mori to be moved here.) Whetstone compares music to sparks of fire and heat and, just a bit later, to Zeuxis’s painting of Cupid. These multisensory descriptions are worth spelling out, strengthening as they would the author’s argument about the interrelated nature of the early modern senses. They also suggest that the Horatian equivalence of poetry/song and image, as explained by Richard Haydock, could be relocated here rather than buried on page 9.

In the introduction and throughout the essay, there are excessive in-text citations of other scholars; only the most important (e.g. Carruthers) should be named by the author in the body of the text. When bits of historical information are brought in, the author can do more to unpack their significance to the argument. For instance, the inclusion of pious sayings (such as Psalms) on the walls and ceilings of homes effectively covered the architecture with chant and song—quite literally giving imparting oral and auditory sense unto the walls (p. 3, ll. 104-07). These wall paintings relate to and invert the essay’s main theme, since they assert the eternity of God’s word over the impermanence of masonry.

As the author moves between the essay’s sections, she should open with topic sentences that better set up the section’s themes and their relationship to the rest of the essay. For instance, in “The Five Senses” section, the author could begin with a sentence akin to “Before entering an in-depth discussion of paintings at Bolsover Castle and Knole House, I want to frame these works within a broader early modern discourse on the five senses and their emergence in country house décor.” On p. 7, l. 245, I would be careful about overstating the functional specialization of early modern domestic spaces; indeed, the multifunctional character of such spaces is what imparts dynamism to their sensory iconography, as they go from hosting feasts and performances to the more quotidian functions of domestic life.

I very much enjoyed the “Mind the Gap” section. Again, however, this section could be better set up. The section title might be renamed “Minding the Sensory Gap at Bolsover Castle.” Currently, the section’s opening sentences are vague and disorienting. They would be better rephrased along the lines of: “In much early modern sensory iconography, the visual imagery is deliberately lacunose, leaving gaps that must literally be filled by the viewer. This is true, for instance, in a set of embroidered mirrors….Whereas the four senses of smell, touch, taste, and hearing depicted with personifications, sight (visus) is missing its attendant human figure….and so forth”

Here, as well as in the following discussion of the Bolsover iconography, the author could better connect such spectral iconography to the essay’s main subject of permanence and impermence, presence and absence. Now the viewer appears, now she disappears; now Cavendish emerges to greet his guests, now he departs, leaving an iconographic vacuum. This dynamic also pertains to the author’s well-placed use of Ben Jonsons’s Bolsover Masque. In the memento mori mode, the singers sighs at the sensory impermanence of both banquets and love (“Would it ever last!” sings the bass; the tenor responds, “We wish the same.”).

The Knole House section is very interesting. Once again, though, the author needs a better set up. Currently, she starts by jumping immediately into the deep end, with a potted history of the residence. A better starting sentence would be something like, “The links between music, architecture, and visual imagery find an arresting elaboration at another great residence, Knole House in Kent.” The author might also consider reformulating the section title, to something like “A Procession of the Senses at Knole House in Kent”

Personally, I’m not totally convinced by the author’s application of “imagination-as-scholarship” in this section. I’m not constitutionally against such creative reconstruction, and its relevance to ephemeral performance, but here it comes out of left field and lacks a clear statement of methodology. The whole imagined scenario, altogether no more than a few paragraphs, feels too brief and non-committal, breaking in and out of the second-person viewpoint. In an alternative essay, the whole project of imaginative reconstruction would be properly thematized right at the beginning and boldly worked throughout the entire essay. But in this context, it feels fragmentary and undercooked. I would eliminate the second-person framing and simply describe the itinerary of “a hypothetical guest.”

The description of the ascent up the staircase is great. There might be more to say about staircases as places of passage, designed to be animated by bodily movement. Like the embroidered mirror, the iconography of vision here is “completed” by the appearance of a seeing user. Note that this iconography relies on a commonplace early modern metaphor of windows as eyes, eyes as windows.

As the author closes this section with hypothetical songs that describing the elusiveness and inconstancy of love, she might draw a larger relationship to the similar themes found in Whetstone’s sonnet and Ben Jonson’s Bolsover Masque. The unreliable quality of the senses that these songs convey, calling into question the five-sense iconography painted on the ceilings and walls, create a flickering quality between materiality and immateriality that can be carried onto the essay’s conclusion.

Author Response

Many thanks for your time and expertise in reviewing my article. I've taken on board as much of your feedback as possible and appreciate your astute observations, which have added valuable insight and certainly strengthened my argument. You've really helped me see my own work in a new light in a few places in particular, so thank you.

First, I must apologise for not having the images embedded in the article, as I never intended to waste anybody's time. I did not submit the article through the portal for peer review myself, as I had originally submitted it to the editors of the Special Issue directly back in November. Eventually, the article was submitted through the Arts Journal portal by someone else on my behalf, and I believe there was some slight miscommunication between the editors surrounding this. Anyhow, I am sorry for the inconvenience this has caused, but I wanted to explain myself as it was not merely laziness or oversight.

While I have decided to leave the title of the article, I have reworked the abstract in line with your suggestions. I also agree that Whetstone's sonnet should have been put to better use, and I have tried to incorporate reference to it in a few additional places.

The opening of the Knole House section is now different (accommodating a request from another reviewer, but also in line with your thoughts). I have also retitled some of the subsections per your recommendations, though some I preferred to leave as they were.

I hope you'll find a clearer statement of methodology in my imagined scenario, and I have unified the multiple viewpoints.

You make an excellent point about the inclusion of pious sayings (such as Psalms) on walls, though I decided against including it here because I felt that it would need an example and would take the paragraph in another direction. Though certainly food for future thought.

I appreciate your recommendation for clearer topic sentences and better organisation overall. Other reviewers thought the article moved seamlessly and clearly between sections and was organised logically as is, so upon further consideration (and in consultation with the special issue editors) I have adjusted only some of the sections where I agree with you that more graceful intros were needed.

Lastly, I agree that I needed to better tie in im/materiality at the conclusion. I drew in Whetstone's sonnet and Bolsover Masque per your suggestion and made reference to what you've called the 'flickering quality' between materiality and immateriality. While it isn't in the overarching conclusion, it is at the end of the song section and makes it much stronger.

Reviewer 3 Report

See attached file

Author Response

I greatly appreciate your time and expertise in reviewing my article. All of your excellent suggestions have been incorporated in my rewrite.

I have tried to make clearer the idea that the experience of 'emotion' and 'aesthetics' as we know them is anachronistic. I've also added a footnote to this effect before the work of Bruce Smith is introduced.

I've also added a footnote to clarify my use of 'mindful awareness' as signifier (rather than any sort of contemporary idea). As you say, it is implied as such, but it can't hurt to make it clearer.

I have drawn more attention to the intellectual circumstances that led me to the article's development in terms of my own social musicking experiences. I agree that the place of experiments to performers is a tradition that needs outlining for those not within musicology. I also hope you don't mind that I have borrowed explicitly from your excellent suggestions in this section (in particular the idea that this type of experimentation has roots in Baconian-style empiricism - great point!).

Reviewer 4 Report

This article makes an excellent, and timely contribution to the field of musical visual culture. The author posits the discussion in visualisations of the five senses in areas where music-making and general recreation occurred, and this cleverly renders the material accessible to a broad disciplinary readership. It will be of immense value to musicologists, art historians and visual/material culture historians alike, and makes strides towards transforming early modern studies in general into a far more interdisciplinary field (far more in keeping with how the arts and knowledge were understood during that period, and following the excellent model set for us early modernists by the medievalists). The analysis of the Grand Staircase as visual object, performance space and silent musical object provides an especially good example of how such scholarship can, and should, be undertaken. I thoroughly recommend publishing this article, subject to a few minor revisions.

The article would be bolstered by referencing some of the older work on sensing in a few places (nothing more than a well-placed footnote or two). It isn’t especially that this is lacking, but rather that it would offer extra assistance for readers who are less familiar with sound studies and music and visual culture interactions. In particular, I’d recommend that David Summers’ Judgment of Sense and E. Ruth Harvey’s Inward Wits, along with some of Constance Classen's work, might provide a good amount of anchorage. Some of the framing material from Niall Atkinson’s Noisy Renaissance might also help here (particularly given that Atkinson is writing from an art history department on sound, so will give readers from visual culture more of a sense of familiarity).

Likewise, for the benefit of readers less familiar with scholarship posited in possibilities rather than the frustratingly scant things that can be said with cold hard fact in this field, it might be worth citing Noemie Ndiaye at some point towards the opening of the article (perhaps after the Wistreich quote). Ndiaye recently and very elegantly defined her research approach for her recent Scripts of Blackness as ‘subjunctive’ - as in, exploring what is imaginable or possible. The way she defines it could be of use to the author in demonstrating that exploring possibility is a legitimate and valuable historical approach, and that it is an important way of engaging with fragmentary or ephemeral historical evidence.

A few small specifics:

At note 46, line 273, page 8 – I suspect it’s Riegl via Gombrich’s influence here (specifically Gombrich’s beholder’s share in Art & Illusion); if you want to get very technical it’s worth having a look at Anil Seth’s ‘From Unconscious Inference to the Beholder’s Share: Predictive Perception and Human Experience’, recently(ish) in the European Review.

At line 216-7, page 6 - I think this can be stated more emphatically. It is well supported in how the article is argued afterwards; it is odd that this interpretation seems to have been enough when the plenitude and variety of the five senses trope in interior decoration suggests that it is far more complex!

At line 101, page 3 – ‘holistically’ for ‘wholistically’.

At line 318, page 9 – a missing ‘the’? Or should it be 'Song at Banquet'?

At line 498, page 14 – worth explicitly mentioning here the relationship between Robert & Richard S.

Author Response

I greatly appreciate your time and expertise in reviewing my article. For the most part, all of your excellent suggestions have been taken into account in my revisions.

I agree that I needed to bolster the citations with additional references to older scholarship on sensing and I have done so accordingly. I wasn't familiar with Noemie Ndiaye's book. I read a few sections and do see how her approaches are of relevance here and I look forward to engaging with this work more in the future (as well as the work of Saidiya Hartman from whom Ndiaye draws).

I ended up cutting the footnote about Emmaus, but I do think you're probably right about the connections to beholder's share/Riegl via Gombrich. I decided against getting technical in the footnote, but I have flagged Anil Seth's article for future reading as this is an idea that I find really interesting and is something to which I suspect I'll return.

In my new draft I have drawn more explicit attention to the sociable musicking (and historiographical antecedents) that have led to my own insights.

Round 2

Reviewer 2 Report

The article is greatly improved after revisions. An overall imaginative text that engages with a substantial body of literature in art/architectural history and musicology. Some stylistic editing might be welcome, but overall the content looks great.